# SLC38A2 provides proline to fulfill unique synthetic demands arising during osteoblast differentiation and bone formation

Leyao Shen[1,2], Yilin Yu[2], Yunji Zhou[3], Shondra M Pruett-Miller[4], Guo-Fang Zhang[5,6], Courtney M Karner[1,2,7]*

[1]Department of Orthopaedic Surgery, Duke University School of Medicine, Durham, United States; [2]Department of Internal Medicine, University of Texas Southwestern Medical Center, Dallas, United States; [3]Department of Biostatistics and Bioinformatics, Duke University School of Medicine, Durham, United States; [4]Department of Cell and Molecular Biology, St. Jude Children's Research Hospital, Memphis, United States; [5]Sarah W. Stedman Nutrition and Metabolism Center & Duke Molecular Physiology Institute, Duke University Medical Center, Durham, United States; [6]Department of Medicine, Duke University School of Medicine, Durham, United States; [7]Charles and Jane Pak Center for Mineral Metabolism and Clinical Research, University of Texas Southwestern Medical Center at Dallas, Dallas, United States

**Abstract** Cellular differentiation is associated with the acquisition of a unique protein signature that is essential to attain the ultimate cellular function and activity of the differentiated cell. This is predicted to result in unique biosynthetic demands that arise during differentiation. Using a bioinformatic approach, we discovered that osteoblast differentiation is associated with increased demand for the amino acid proline. When compared to other differentiated cells, osteoblast-associated proteins, including RUNX2, OSX, OCN, and COL1A1, are significantly enriched in proline. Using a genetic and metabolomic approach, we demonstrate that the neutral amino acid transporter SLC38A2 acts cell-autonomously to provide proline to facilitate the efficient synthesis of proline-rich osteoblast proteins. Genetic ablation of SLC38A2 in osteoblasts limits both osteoblast differentiation and bone formation in mice. Mechanistically, proline is primarily incorporated into nascent protein with little metabolism observed. Collectively, these data highlight a requirement for proline in fulfilling the unique biosynthetic requirements that arise during osteoblast differentiation and bone formation.

## Editor's evaluation

The article is novel and informative; the authors' conclusions are supported by the data as shown. The article is significant as it proves that a key function of proline during bone formation is being incorporated into proline-enriched proteins rather than contributing to other metabolic processes.

*For correspondence:
courtney.karner@
utsouthwestern.edu

Competing interest: The authors declare that no competing interests exist.

**eLife digest** Bones have diverse roles in the body, such as supporting weight, allowing movement and protecting internal organs. Regardless of their location, all bones in the body are formed and maintained by specialized cells called osteoblasts. To produce the different components of bone, osteoblasts need a constant supply of amino acids, the building blocks of proteins. If these nutrients are limited, this may lead to weak and fragile bones that can fracture more easily.

Naïve cells in the bone, which are yet to have a defined role, also require large amounts of amino acids to develop into fully functioning osteoblasts. Previous studies have found that specific amino acids (like glutamine and asparagine) are particularly important for forming the proteins in bone. However, it was unclear which amino acids are critical for the development of osteoblasts.

To investigate, Shen et al. studied naïve cells that had been extracted from the embryos of mice and developed into osteoblasts in the laboratory. They found that the developing osteoblasts produced proteins enriched in proline, and that naïve cells required large amounts of this amino acid as they turned into osteoblasts.

Genetic analysis revealed that osteoblasts carry the gene for a protein called SLC38A2, which has been shown to transport proline into other types of cells. Shen et al. then used gene editing tools to delete this transporter from the osteoblasts of mice. The mutated mice could not efficiently produce proline-rich proteins during embryonic development and formed less bone.

These findings highlight that proline is important for developing osteoblasts and synthesizing the products of bone. Further research is needed, but it is possible that dietary supplements of proline may be beneficial for maintaining or promoting bone formation in adulthood. This could help individuals that have more fragile bones, such as the elderly or patients with bone diseases, like osteoporosis.

## Introduction

The mammalian boney skeleton is a remarkable organ that has multiple functions, including support, mobility, protection of internal organs, endocrine signaling, mineral storage, as well as being a site for red blood cell production (*Guntur and Rosen, 2012*; *Jagannathan-Bogdan and Zon, 2013*; *Long, 2011*; *Salhotra et al., 2020*). The skeleton develops embryonically through two distinct mechanisms, intramembranous and endochondral ossification (*Berendsen and Olsen, 2015*). Intramembranous ossification is responsible for forming the 'flat' bones of the skull. Here, mesenchymal progenitor cells condense and give rise to bone directly. The remainder of the skeleton develops through endochondral ossification. In this process, the mesenchymal progenitors condense and give rise to a cartilaginous template that is subsequently ossified. Regardless of the developmental mechanism, skeletal development depends upon osteoblasts. Osteoblasts are secretory cells responsible for producing and secreting the collagen type 1 (COL1A1)-rich extracellular bone matrix. Osteoblast differentiation is tightly regulated by the transcription factors RUNX2 and OSX (encoded by *Sp7*) (*Ducy et al., 1997*; *Nakashima et al., 2002*; *Otto et al., 1997*; *Takarada et al., 2016*). Genetic studies in mice demonstrate that RUNX2 is essential for commitment to the osteoblast lineage as well as the transcriptional regulation of osteoblast marker genes (e.g., *Spp1* and *Bglap*) (*Komori et al., 1997*; *Meyer et al., 2014*; *Otto et al., 1997*; *Wu et al., 2014*). OSX functions downstream of RUNX2 to regulate osteoblast differentiation and osteoblast gene expression (e.g., *Spp1*, *Ibsp*, and *Bglap*) (*Bianco et al., 1991*; *Ducy et al., 1996*).

During differentiation, osteoblasts acquire a distinct protein profile in addition to increasing bone matrix production (*Alves et al., 2010*; *Zhang et al., 2007*). Protein and bone matrix production is biosynthetically demanding and predicted to present differentiating osteoblasts with changing metabolic demands (*Buttgereit and Brand, 1995*). Thus, osteoblasts must maximize nutrient and amino acid acquisition for differentiation and matrix production to proceed. Consistent with this, both glucose and amino acid uptake are required for osteoblast differentiation and bone formation (*Elefteriou et al., 2006*; *Rached et al., 2010*; *Wei et al., 2015*). Osteoblasts primarily rely on glycolytic metabolism of glucose that provides ATP for protein synthesis and to regulate RUNX2 stability to promote osteoblast differentiation (*Esen et al., 2013*; *Lee et al., 2020*; *Wei et al., 2015*). Like glucose, amino acids have long been recognized as important regulators of osteoblast differentiation and bone matrix production (*Elefteriou et al., 2006*; *Hahn et al., 1971*; *Karner et al., 2015*; *Rached*

*et al., 2010*; *Shen and Karner, 2021*; *Yu et al., 2019*). Affecting the ability of cells to sense or obtain amino acids either by limiting their availability in the media or inhibiting cellular uptake has detrimental effects on osteoblast differentiation and bone formation (*Chen and Long, 2018*; *Elefteriou et al., 2006*; *Esen et al., 2013*; *Hu et al., 2020*; *Karner et al., 2015*; *Rached et al., 2010*; *Shen et al., 2021*; *Yu et al., 2019*). Despite this, the role of individual amino acids in osteoblasts is not well understood. Recent studies identified glutamine as a particularly important amino acid in osteoblasts supporting protein and amino acid synthesis, redox regulation, and energetics (*Karner et al., 2015*; *Shen et al., 2021*; *Stegen et al., 2021*; *Yu et al., 2019*). Whether other individual amino acids are similarly important for osteoblast differentiation remains unknown.

Proline is an intriguing amino acid in osteoblasts as it is important for both the biosynthesis and structure of collagen (*Grant and Prockop, 1972*; *Krane, 2008*). In addition, interest in proline has recently increased as proline is critical for cancer cell survival, tumorigenesis, and metastasis (*Elia et al., 2017*; *Liu et al., 2012a*; *Nagano et al., 2017*; *Olivares et al., 2017*; Phang, Liu, Hancock, & Christian, 2012). Proline is a multifunctional amino acid with important roles in carbon and nitrogen metabolism, oxidative stress protection, cell signaling, nutrient adaptation, and cell survival (*Hollinshead et al., 2018*; *Liu et al., 2012c*; *Phang, 2019*). Proline can contribute to protein synthesis directly through incorporation into protein or can be metabolized into downstream products involved in energetic and biosynthetic reactions. Despite its emerging role in cancer cells, the role of proline during osteoblast differentiation and bone development is understudied.

Here, we identify proline as a critical nutrient in osteoblasts. Using a multifaceted approach, we demonstrate that sodium-dependent neutral amino acid transporter-2 (SNAT2, encoded by *Slc38a2* and denoted herein as SLC38A2) acts cell-autonomously to provide proline necessary for osteoblast differentiation and bone development. Mechanistically, proline is essential for the synthesis of proline-rich osteoblast proteins, including those that regulate osteoblast differentiation (e.g., RUNX2 and OSX) and bone matrix production (e.g., COL1A1). These data highlight a broad requirement for proline to fulfill unique synthetic demands associated with osteoblast differentiation and bone formation.

## Results

### Proline is enriched in osteoblast-associated proteins, leading to increased proline demand during osteoblast differentiation

To identify if there are unique requirements for individual amino acids that arise during differentiation, we first profiled the amino acid composition of select proteins (e.g., RUNX2, OSX, COL1A1, and OCN) that are induced during osteoblast differentiation (*Figure 1—figure supplement 1*). These classical osteoblast proteins are enriched with the amino acid proline and to a lesser extent alanine when compared to all proteins (*Figure 1A*, *Table 1*). For comparison, other amino acids were either uniformly underrepresented (e.g., Glu, Ile, and Val) or were enriched only in a subset of these proteins (e.g., Cys, Gly, and Gln) (*Figure 1A*, *Table 1*). To determine if this observation was characteristic of osteoblast proteins in general, we next evaluated amino acid enrichment in proteins that are associated with osteoblast differentiation based on Gene Ontology (GO). Osteoblast-associated proteins (GO:0001649) were found to have a higher proline composition when compared to the average of all proteins (7.1% vs. 6.1% proline for osteoblasts vs. all proteins) (*Table 2*). In fact, many classical osteoblast proteins (e.g., RUNX2, OSX, and COL1A1) were above the 90th percentile for proline composition and 43.5% of all osteoblast proteins were above the 75th percentile for proline composition. No other amino acids were similarly enriched in osteoblast-associated proteins (*Table 2*). Moreover, osteoblast-associated proteins were enriched for proline when compared to proteins associated with other cell types, including osteoclasts (GO:0030855), cardiomyocytes (GO:0001649), muscle cells (GO:0055007), hematopoietic stem cells (GO:0042692), endothelial cells (GO:0030182), epithelial cells (GO:0055007), or neurons (GO:0030182) (*Figure 1B*, *Table 2*). In contrast, alanine and cysteine enrichment was comparable amongst the different cell types (*Figure 1B*, *Table 2*). These data suggest that osteoblast differentiation is associated with increased proline demand. To test this hypothesis, we transcriptionally profiled naïve calvarial cells that were induced to undergo osteoblast differentiation and calculated the proline enrichment of the encoded proteins. Consistent with our previous analysis, proline was enriched in proteins encoded by the induced genes compared to either all genes or genes that were suppressed in differentiated calvarial osteoblasts (*Figure 1C*). Moreover, comparing

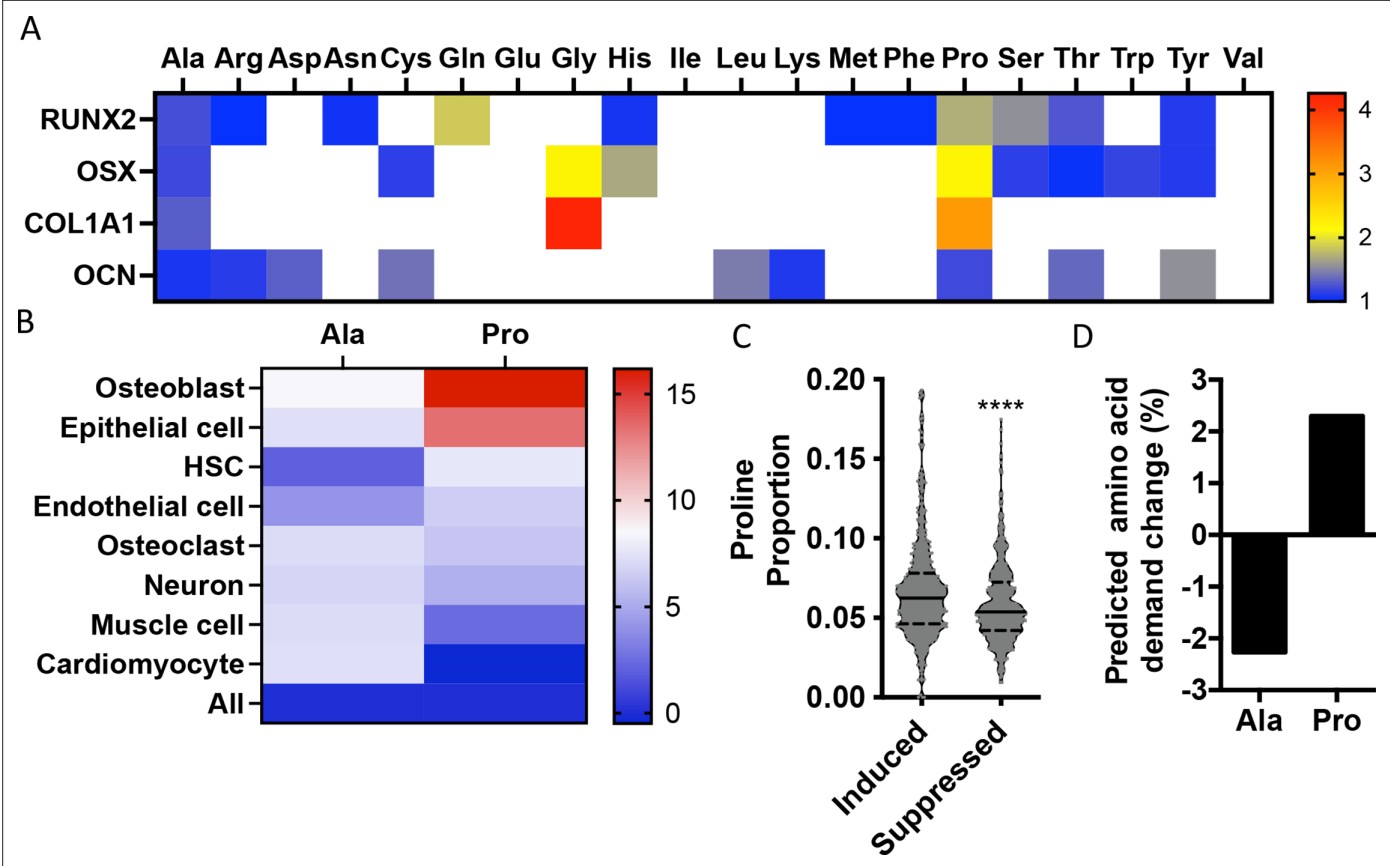

**Figure 1.** Osteoblast proteins are enriched with the amino acid proline. (**A**) Heat map depicting the relative amino acid enrichment for the indicated osteoblast proteins. Color bar represents fold enrichment relative to the average amino acid content. White boxes denote below-average enrichment. (**B**) Heat map depicting alanine or proline enrichment in differentiation-associated proteins. Color bar represents the percent increase in abundance relative to all proteins. (**C**) Volcano plot depicting the proline proportion of the top 500 genes that are induced or suppressed during osteoblast differentiation. Dashed lines denote quartiles while the solid line denotes the median. ****$p \leq 0.00005$ by unpaired two-tailed Student's *t*-test. (**D**) Graphical depiction of the predicted change in demand for alanine or proline based on changes in gene expression during osteoblast differentiation. See numerical source data and RNAseq source data in *Figure 1—source data 1*.

The online version of this article includes the following source data and figure supplement(s) for figure 1:

**Source data 1.** Numerical source data for *Figure 1*.

**Figure supplement 1.** Predicted amino acid demand changes during osteoblast differentiation.

**Figure supplement 1—source data 1.** Numerical source data for *Figure 1—figure supplement 1*.

the basal and differentiation-associated transcriptional changes with proline composition indicates that proline demand is predicted to rise, whereas the demand for alanine and cysteine is predicted to decline during osteoblast differentiation (*Figure 1D*, *Figure 1—figure supplement 1C*). Altogether, these data predict that proline is uniquely required during osteoblast differentiation due to the increased expression of proline-enriched osteoblast proteins.

We next sought to understand proline dynamics in osteoblasts. Proline can be taken up from the extracellular milieu or synthesized. To determine the source of proline in osteoblasts, we first performed stable isotopomer analysis using $^{13}C_U$-proline to evaluate proline uptake or either $^{13}C_U$-glutamine or $^{13}C_{1,2}$-glucose to estimate de novo proline biosynthesis. 10.5% of intracellular proline is synthesized from either glutamine (9.9%) or glucose (0.6%) in 24 hr (*Figure 2A*). By comparison, 37.8% of the proline pool is labeled from $^{13}C_U$-proline after 24 hr, and this increased to 66.6% after 72 hr (*Figure 2A*). The slow labeling of proline when compared to intracellular glutamine that reached saturation within hours suggests that in naïve calvarial cells proline uptake is slow and the intracellular proline pool is relatively stable with little turnover. To test this, we performed radiolabeled amino acid uptake assays to compare the rates of proline and glutamine uptake. Consistent with the labeling

**Table 1.** Amino acid composition of classical osteoblast proteins.

| | RUNX2 | OSX | COL1A1 | OCN | All proteins |
|---|---|---|---|---|---|
| Ala | 0.084 | 0.082 | 0.089 | 0.074 | 0.068 |
| Cys | 0.012 | 0.026 | 0.012 | 0.032 | 0.023 |
| Asp | 0.044 | 0.033 | 0.041 | 0.063 | 0.048 |
| Glu | 0.021 | 0.044 | 0.052 | 0.063 | 0.069 |
| Phe | 0.038 | 0.023 | 0.018 | 0.021 | 0.038 |
| Gly | 0.053 | 0.138 | 0.268 | 0.053 | 0.063 |
| His | 0.028 | 0.044 | 0.006 | 0.000 | 0.026 |
| Ile | 0.021 | 0.012 | 0.017 | 0.042 | 0.045 |
| Lys | 0.031 | 0.051 | 0.038 | 0.063 | 0.057 |
| Leu | 0.059 | 0.084 | 0.035 | 0.147 | 0.100 |
| Met | 0.023 | 0.012 | 0.010 | 0.021 | 0.023 |
| Asn | 0.038 | 0.026 | 0.023 | 0.032 | 0.036 |
| Pro | 0.105 | 0.133 | 0.190 | 0.074 | 0.061 |
| Gln | 0.089 | 0.033 | 0.033 | 0.032 | 0.048 |
| Arg | 0.056 | 0.040 | 0.047 | 0.063 | 0.056 |
| Ser | 0.133 | 0.098 | 0.046 | 0.074 | 0.085 |
| Thr | 0.069 | 0.056 | 0.030 | 0.074 | 0.054 |
| Val | 0.054 | 0.021 | 0.029 | 0.032 | 0.061 |
| Trp | 0.010 | 0.014 | 0.004 | 0.000 | 0.012 |
| Tyr | 0.030 | 0.030 | 0.010 | 0.042 | 0.027 |

data, proline uptake was slow compared to glutamine uptake in naïve cells (*Figure 2B*, *Figure 2—figure supplement 1A*). During differentiation, the rate of proline uptake increased significantly and to a greater extent than glutamine, which also increased in both primary bone marrow stromal cell and calvarial cultures (*Figure 2C*, *Figure 2—figure supplement 1B*). By comparison, the uptake of alanine was unchanged during differentiation (*Figure 2—figure supplement 1B*). The tracing experiments indicated that little proline metabolism occurs in osteoblasts as proline carbon was not observed in glutamate or any other amino acid or other downstream metabolites (e.g., α–ketoglutarate, malate, or citrate) even after 72 hr (*Figure 2A*, *Figure 2—figure supplement 1C*). By comparison, carbon from both glutamine and glucose was observed in many metabolites, including α–ketoglutarate, malate, citrate, and various amino acids (*Figure 2A*, *Figure 2—figure supplement 1C*). These data suggest that proline is not metabolized to glutamate or other downstream metabolites. Rather, proline is primarily used for protein synthesis. Consistent with this conclusion, proline incorporation into both total protein and collagen significantly increases during differentiation (*Figure 2D*). Moreover, almost 50% of the proline in total protein was derived from $^{13}C_U$-proline (*Figure 2E*). Importantly, we observed no proline-derived amino acids in total protein despite the presence of glutamine-derived amino acids including proline (*Figure 2E*). Thus, proline demand and protein synthesis rise concomitantly during osteoblast differentiation.

We next sought to determine the effects of proline withdrawal on protein expression. Proline withdrawal specifically reduced charging of the proline tRNA (AGG) but did not affect the activation of either the mTOR pathway (as determined by S6 ribosomal protein phosphorylation at S235/236) or the integrated stress response (ISR) (as determined by EIF2α phosphorylation at Ser51) (*Figure 3—figure supplement 1A and B*). Proline withdrawal did not affect the expression of select non-proline-enriched proteins (*Figure 3A*). Conversely, proline withdrawal significantly reduced the expression of osteoblast proteins that had higher-than-average proline content, including COL1A1 (19.1% proline),

**Table 2.** Relative amino acid composition of proteins associated with various differentiated cell types based on Gene Ontology (GO) terms.

| | Osteoblast | Epithelial cell | Hematopoietic stem cell | Endothelial cell | Osteoclast | Neuron | Muscle cell | Cardiomyocyte | All |
|---|---|---|---|---|---|---|---|---|---|
| GO term | 00001649 | 0030855 | 0030097 | 0045446 | 0030316 | 0030182 | 0042692 | 0055007 | |
| Ala | 0.0739 | 0.0731 | 0.0695 | 0.0709 | 0.0730 | 0.0728 | 0.0730 | 0.0731 | 0.0681 |
| Cys | 0.0301 | 0.0230 | 0.0234 | 0.0259 | 0.0292 | 0.0228 | 0.0233 | 0.0223 | 0.0227 |
| Asp | 0.0478 | 0.0483 | 0.0493 | 0.0467 | 0.0468 | 0.0502 | 0.0511 | 0.0490 | 0.0479 |
| Glu | 0.0639 | 0.0678 | 0.0671 | 0.0682 | 0.0614 | 0.0693 | 0.0727 | 0.0690 | 0.0694 |
| Phe | 0.0331 | 0.0331 | 0.0365 | 0.0342 | 0.0374 | 0.0347 | 0.0367 | 0.0373 | 0.0375 |
| Gly | 0.0686 | 0.0698 | 0.0672 | 0.0685 | 0.0656 | 0.0667 | 0.0666 | 0.0667 | 0.0629 |
| His | 0.0273 | 0.0256 | 0.0266 | 0.0239 | 0.0245 | 0.0254 | 0.0249 | 0.0256 | 0.0262 |
| Ile | 0.0362 | 0.0401 | 0.0409 | 0.0458 | 0.0407 | 0.0420 | 0.0429 | 0.0441 | 0.0445 |
| Lys | 0.0531 | 0.0557 | 0.0570 | 0.0531 | 0.0499 | 0.0560 | 0.0606 | 0.0627 | 0.0571 |
| Leu | 0.0946 | 0.0923 | 0.0957 | 0.0950 | 0.1040 | 0.0961 | 0.0930 | 0.0903 | 0.1004 |
| Met | 0.0220 | 0.0232 | 0.0229 | 0.0224 | 0.0210 | 0.0222 | 0.0222 | 0.0238 | 0.0228 |
| Asn | 0.0360 | 0.0361 | 0.0362 | 0.0382 | 0.0366 | 0.0374 | 0.0369 | 0.0363 | 0.0360 |
| Pro | 0.0711 | 0.0694 | 0.0659 | 0.0652 | 0.0650 | 0.0644 | 0.0627 | 0.0609 | 0.0612 |
| Gln | 0.0469 | 0.0472 | 0.0468 | 0.0458 | 0.0446 | 0.0459 | 0.0464 | 0.0482 | 0.0478 |
| Arg | 0.0607 | 0.0573 | 0.0563 | 0.0566 | 0.0539 | 0.0575 | 0.0561 | 0.0547 | 0.0559 |
| Ser | 0.0853 | 0.0874 | 0.0848 | 0.0807 | 0.0843 | 0.0835 | 0.0797 | 0.0848 | 0.0853 |
| Thr | 0.0527 | 0.0529 | 0.0537 | 0.0557 | 0.0563 | 0.0538 | 0.0528 | 0.0544 | 0.0543 |
| Val | 0.0568 | 0.0588 | 0.0599 | 0.0639 | 0.0623 | 0.0598 | 0.0599 | 0.0585 | 0.0610 |
| Trp | 0.0119 | 0.0110 | 0.0122 | 0.0119 | 0.0135 | 0.0117 | 0.0108 | 0.0105 | 0.0119 |
| Tyr | 0.0278 | 0.0279 | 0.0280 | 0.0275 | 0.0298 | 0.0279 | 0.0275 | 0.0278 | 0.0270 |

RUNX2 (10.5% proline), OSX (13.3% proline), and ATF4 (10.6% proline) (*Figure 3A*). Importantly, proline withdrawal did not affect the mRNA expression of these proteins (*Figure 3—figure supplement 1C*). We next took a candidate approach and evaluated other proline-enriched (e.g., EIF4EBP1 [13.7% proline], PAX1 [11.1% proline], ATF2 [10.7% proline], SMAD1 [9.9% proline], and EIF2A [7.6% proline]) and non-enriched proteins (ERK1 [6.6% proline], PHGDH [5.3% proline], EEF2 [5.2% proline], AKT [4.6% proline], ACTB [5.1% proline], mTOR [4.4% proline], TUBA [4.4% proline], and S6RP [4.2% proline]) that are not known to be required for osteoblast differentiation but are expressed in calvarial cells according to our transcriptomic analyses. Proline withdrawal significantly reduced the expression of the proline-enriched proteins without affecting the low proline proteins (*Figure 3A*). The reduction in protein expression significantly correlated with the proline content in the proteins (*Figure 3B*). These data indicate that this phenomenon is broadly generalizable in osteoblasts. The decreased protein expression is due primarily to reduced synthesis of proline-enriched proteins as cycloheximide (CHX) washout experiments found that proline withdrawal resulted in a significant delay in the recovery of proline-enriched protein expression (*Figure 3C–G*, *Figure 3—figure supplement 1D–F*). Thus, proline is essential for the synthesis of proline-enriched osteoblast proteins.

## SLC38A2 provides proline to facilitate the synthesis of proline-rich osteoblast proteins

We next sought to identify the proline transporter in osteoblasts. Proline uptake in osteoblasts is reported to occur in a 2-(methylamino)-isobutyric acid (MeAIB)-sensitive manner (*Baum and Shteyer,*

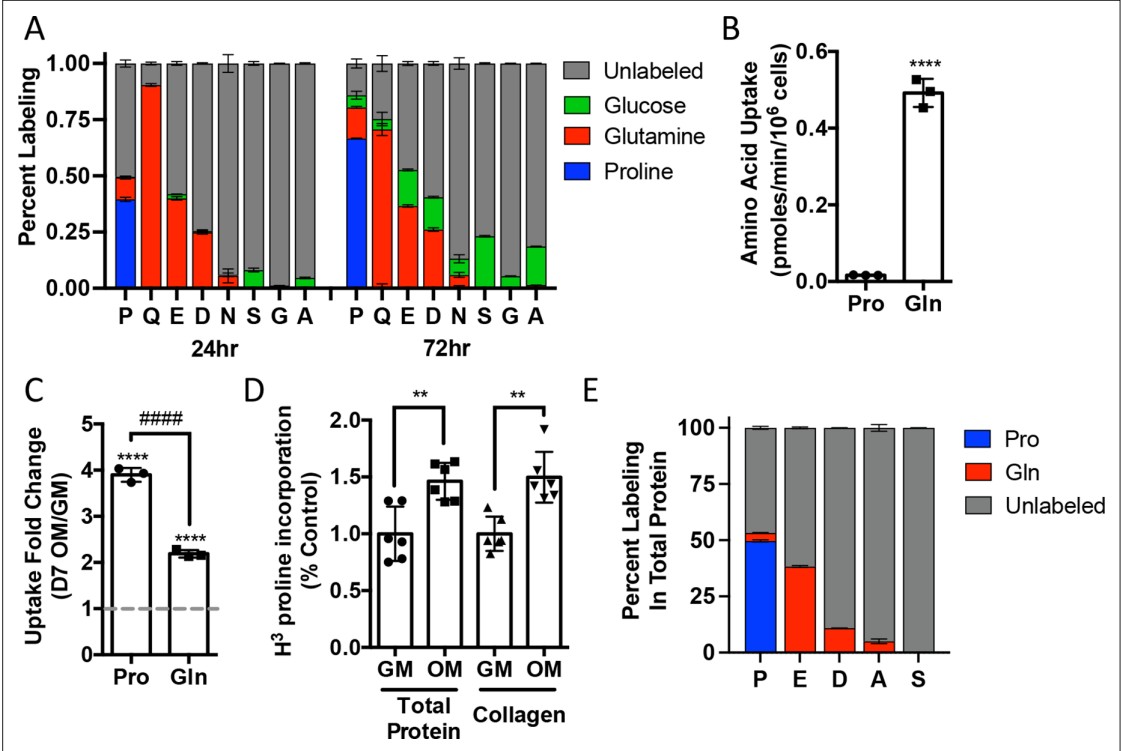

**Figure 2.** Proline uptake and incorporation into protein increases during osteoblast differentiation. (**A**) Graphical depiction of proline, glutamine, glutamate, aspartate, asparagine, serine, glycine, and alanine labeling from [U-¹³C]-proline (n = 3), [U-¹³C]-glutamine (n = 3), or [1,2-¹³C]-glucose (n = 3) in naïve calvarial osteoblasts. (**B, C**) Radiolabeled ³H-proline uptake assay performed in naive bone marrow stromal cells (BMSC) (n = 3) (**B**) or after 7 days of osteoblast differentiation (n = 3) (**C**). ****p≤0.00005 for osteogenic media (OM) vs. growth media (GM, denoted by dashed line), ####p≤0.00005 comparison between change in proline and glutamine by unpaired two-tailed Student's *t*-test. (**D**) Radiolabeled proline incorporation assay performed in primary calvarial cells cultured in GM or OM for 7 days (n = 6). **p≤0.005 by unpaired two-tailed Student's *t*-test. (**E**) Contribution of [U-¹³C]-proline or [U-¹³C]-glutamine to proline, glutamate, aspartate, alanine, or serine isolated from total protein (n = 3). See numerical source data and isotopomer-tracing source data in *Figure 2—source data 1*.

The online version of this article includes the following source data and figure supplement(s) for figure 2:

**Source data 1.** Numerical source data for *Figure 2*.

**Figure supplement 1.** Proline uptake increases during osteoblast differentiation and does not contribute to TCA cycle intermediates.

**Figure supplement 1—source data 1.** Numerical source data for *Figure 2—figure supplement 1*.

*1987*; *Yee, 1988*). Consistent with these reports, MeAIB reduced proline uptake by 80% in both osteoblasts and bone shafts with minimal effects on the uptake of other amino acids (e.g., Gln, Ala, Gly, or Ser) (*Figure 4A*, *Figure 4—figure supplement 1A*). We next sought to identify candidate proline transporters based on relative mRNA expression. Evaluation of our transcriptomic data identified *Slc38a2* as the highest expressed putative proline transporter in calvarial cells (*Table 3*). *Slc38a2* encodes the sodium-dependent neutral amino acid transporter-2 (SNAT2, denoted here as SLC38A2), which transports neutral alpha amino acids (e.g., proline) in a Na⁺-dependent manner that is sensitive to MeAIB (*Grewal et al., 2009*; *Hoffmann et al., 2018*). To determine if SLC38A2 transports proline in differentiating osteoblasts, we targeted *Slc38a2* using a CRISPR/Cas9 approach (*Figure 4—figure supplement 1B*). *Slc38a2* targeting significantly reduced SLC38A2 protein and reduced radiolabeled proline uptake by ~50% in differentiated calvarial cells (*Figure 4B and C*). This is likely a slight underestimation of SLC38A2-dependent proline uptake due to incomplete ablation of SLC38A2 protein (*Figure 4C*). Next, we evaluated the effects of *Slc38a2* ablation on intracellular amino acid concentration using mass spectrometry. Proline and glutamine were both reduced, whereas no other amino acid was significantly affected by the loss of SLC38A2 (*Figure 4—figure supplement 1C*). The reduction in glutamine concentration is attributed to a compensatory increase in de novo proline synthesis from glutamine in the SLC38A2-deficient cells (*Figure 4—figure supplement 1D*). Consistent with

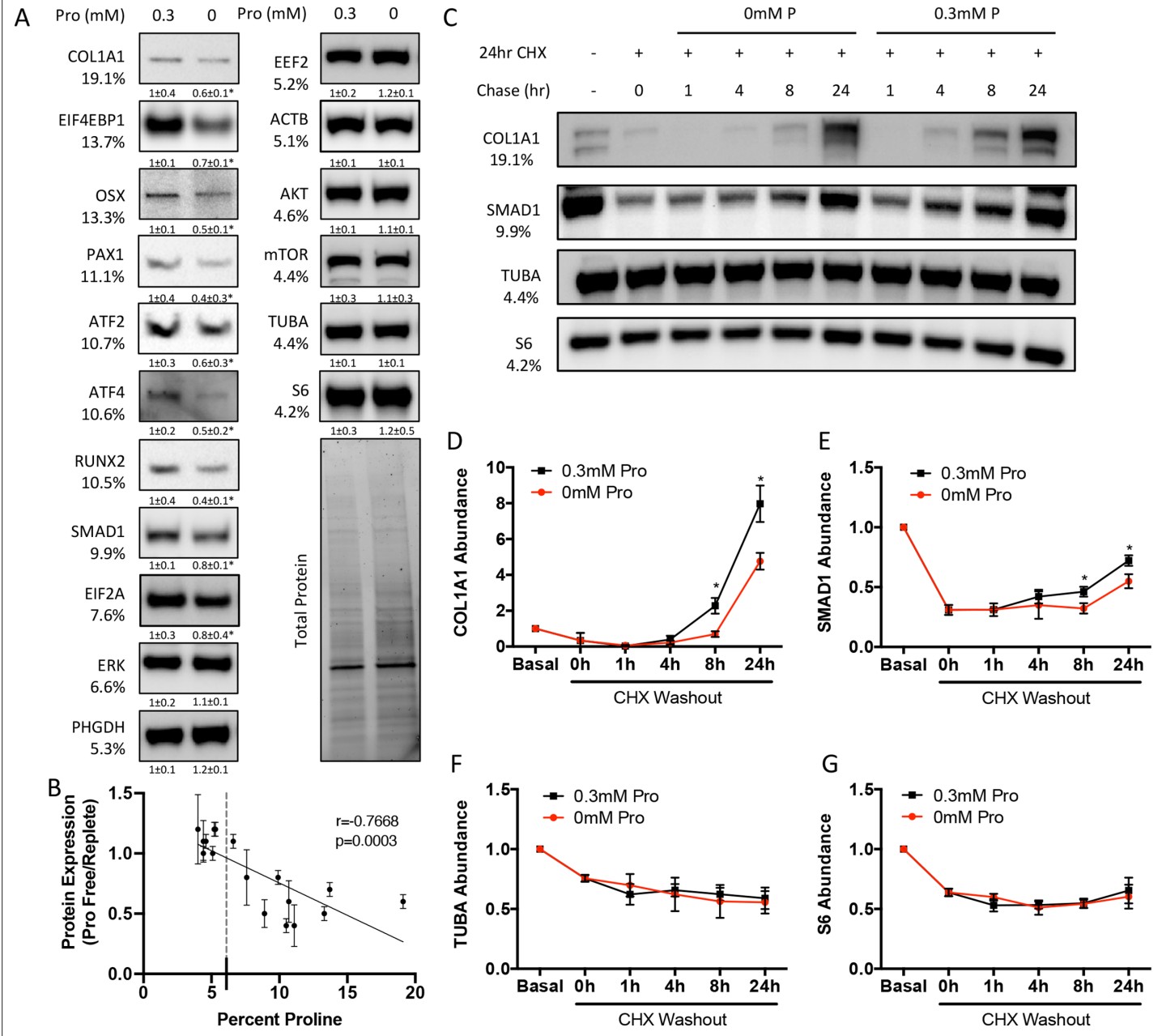

**Figure 3.** Proline is essential for the synthesis of proline-enriched osteoblast proteins. (**A**) Western blot analyses of naïve calvarial cells cultured in 0.3 mM or 0 mM proline for 48 hr (n = 3). In all blots, the percent proline composition is noted under the protein name. Protein expression normalized to total protein. Fold change ± SD for three independent experiments. (**B**) Correlation analysis of protein expression as a function of the proline composition of proteins in naïve calvarial cells cultured in media containing either 0 mM or 0.3 mM proline for 48 hr. (**C–G**) The effect of proline availability on the synthesis of select proteins (n = 3). CHX, cycloheximide. Error bars depict SD. *p≤0.05 by unpaired two-tailed Student's *t*-test. See numerical source data and uncropped Western blot images in *Figure 3—source data 1*.

The online version of this article includes the following source data and figure supplement(s) for figure 3:

**Source data 1.** Numerical and uncropped western blot source data for *Figure 3*.

**Figure supplement 1.** Proline is essential for the synthesis of proline-enriched osteoblast proteins.

**Figure supplement 1—source data 1.** Numerical and uncropped western blot source data for *Figure 3—figure supplement 1*.

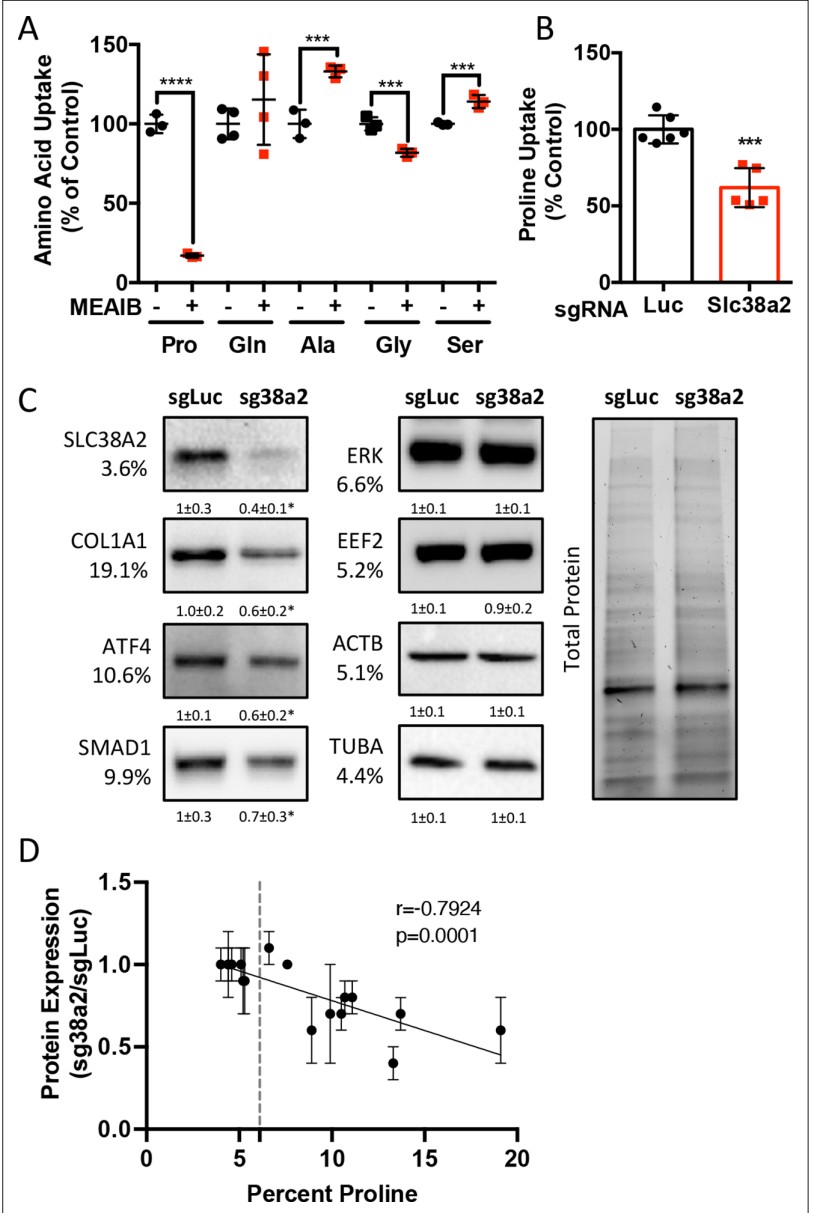

**Figure 4.** *Slc38a2* provides proline critical for the synthesis of proline-rich proteins. (**A**) Graphical depiction of the effects of 5 mM 2-(methylamino)-isobutyric acid (MeAIB) on radiolabeled amino acid uptake in primary calvarial cells (n = 3; except for glutamine n = 4). (**B, C**) Effect of *Slc38a2* targeting on $^3$H-proline uptake (sgLuc n = 6; sgSlc38a2 n = 5) (**B**), or protein expression (n = 3) (**C**). In all blots, the percent proline composition is noted under the protein name. Protein expression normalized to total protein. Fold change ± SD for three independent experiments. (**D**) Correlation analysis of protein expression as a function of the proline composition of proteins in *Slc38a2* (sg38a2) targeted or control (sgLuc) calvarial cells. *p≤0.05, ***p≤0.0005, ****p≤0.00005 by unpaired two-tailed Student's *t*-test. See numerical source data and uncropped Western blot images in *Figure 4—source data 1*.

The online version of this article includes the following source data and figure supplement(s) for figure 4:

**Source data 1.** Numerical and uncropped western blot source data for *Figure 4*.

**Figure supplement 1.** *Slc38a2* provides proline critical for the synthesis of proline-rich proteins.

**Figure supplement 1—source data 1.** Numerical and uncropped western blot source data for *Figure 4—figure supplement 1*.

**Table 3.** mRNA expression of putative proline transporters.

|  | System | Alias | cOB |
|---|---|---|---|
|  |  |  | FPKM |
| Slc38a2 | A | SNAT2 | 8823.1 |
| Slc1a4 | ASC | ASCT1 | 3030.7 |
| Slc36a4 | LYAAT | PAT4 | 1929 |
| Slc36a1 | LYAAT | PAT1 | 803.7 |
| Slc38a4 | A | SNAT4 | 267.9 |
| Slc36a2 | LYAAT | PAT2 | 1.1 |
| Slc6a15 | B[0] | B[0]AT2 | 7.5 |
| Slc36a3 | LYAAT | PAT3 | 0 |
| Slc6a7 | IMINO[B] | PROT | 4.3 |
| Slc6a20a | IMINO | SIT2 | 0 |
| Slc6a20b | IMINO | SIT1 | 0 |
| Slc6a19 | B[0] | B[0]AT1 | 0 |

decreased proline uptake, *Slc38a2* ablation specifically reduced proline-tRNA charging similar to proline withdrawal without negatively affecting charging of other tRNAs or activating the ISR (*Figure 4—figure supplement 1E and F*). Moreover, *Slc38a2* ablation specifically reduced the expression of the proline-enriched proteins without affecting the expression of non-proline-enriched proteins or the mRNA expression of these proteins (*Figure 4C*, *Figure 4—figure supplement 1F and G*). The effect of *Slc38a2* ablation on protein expression significantly correlated with the proline content in the proteins (*Figure 4D*). Importantly, *Slc38a2*-deficient cells were characterized by significantly reduced rates of collagen synthesis and protein synthesis in general (*Figure 4—figure supplement 1H and I*). This is likely a direct result of decreased proline uptake as *Slc38a2* deletion did not affect either mTOR activation or induce ISR (*Figure 4—figure supplement 1F*). Thus, SLC38A2 provides proline for the efficient synthesis of proline-enriched proteins.

## *SLC38A2* provides proline necessary for bone development

We next sought to understand the role of SLC38A2 during osteoblast differentiation. *Slc38a2* deletion did not affect cell viability, proliferation, or the mRNA expression or induction of early osteoblast regulatory genes (e.g., *Akp2* and *Runx2*) (*Figure 5—figure supplement 1A–C*). However, *Slc38a2*-deficient cells were characterized by reduced induction of *Sp7* and terminal osteoblast marker genes (e.g., *Ibsp* and *Bglap*) as well as reduced matrix mineralization (*Figure 5—figure supplement 1C and D*). This indicates that SLC38A2 provides proline required for terminal osteoblast differentiation and matrix mineralization in vitro.

In light of these data, we next analyzed the function of *Slc38a2* during osteoblast differentiation by comparing mice null for SLC38A2 due to the insertion of LacZ into the coding region of *Slc38a2* (*Slc38a2^LacZ/LacZ*). We verified the absence of SLC38A2 expression by Western blot (*Figure 5—figure supplement 2A*). Using Alcian blue and Alizarin red staining (which stains cartilage and bone matrix blue or red, respectively), we found that *Slc38a2^LacZ/LacZ* embryos were characterized by a conspicuous reduction in red mineralized bone matrix staining in both endochondral and intramembranous bones at embryonic day (E)15.5 (*Figure 5A and B*). This defect in bone mineralization was most obvious in the developing skull (*Figure 5B*). By comparison, *Slc38a2^LacZ/LacZ* animals had no apparent defects in cartilage formation at E15.5, indicating that loss of Slc38a2 impacts osteoblast differentiation. To test this, we crossed mice harboring a floxed allele of *Slc38a2* (*Slc38a2^fl*) with mice expressing Cre recombinase under the control of the *Sp7* promoter (*Sp7Cre*), which is active in osteoblast progenitors beginning at E14.5 (*Rodda and McMahon, 2006*). *Sp7Cre;Slc38a2^fl/fl* bones were characterized by reduced SLC38A2 expression and reduced proline uptake (*Figure 5—figure supplement 3A and B*). Like the *Slc38a2^LacZ/LacZ* mice, *Sp7Cre;Slc38a2^fl/fl* mice had significantly less Alizarin red-stained bone matrix at E15.5 (*Figure 5C and D*). By postnatal day 1 (P1), overall bone matrix in long bones was comparable in both genetic models; however, the skulls from both *Slc38a2^LacZ/LacZ* and *Sp7Cre;Slc38a2^fl/fl* mice continued to be poorly mineralized with patent fontanelles compared to their respective littermate controls (*Figure 5E–H*, *Figure 5—figure supplement 3C*).

*Sp7Cre* is expressed in both osteoblasts and hypertrophic chondrocytes in the developing limbs. During the characterization of these mice, we observed a significant delay in COLX removal at E15.5 in *Sp7Cre;Slc38a2^fl/fl* mice (*Figure 5—figure supplement 3D*). This is consistent with delayed endochondral ossification and suggests SLC38A2 may also function in chondrocytes. Because of this and

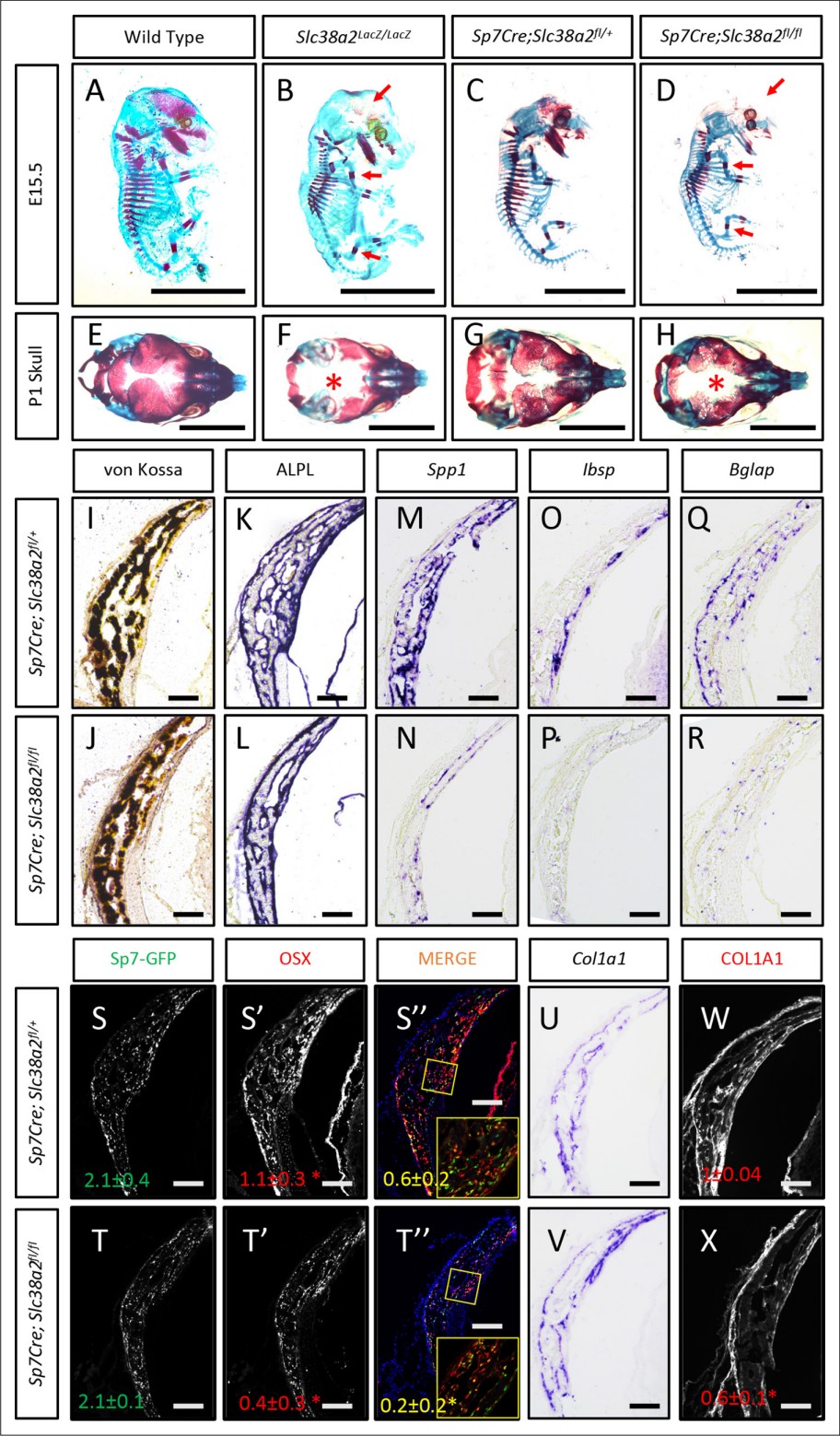

**Figure 5.** *Slc38a2*-dependent proline uptake is required for osteoblast differentiation during bone development. (**A–H**) Skeletal preparations of *Slc38a2^LacZ/LacZ^* or wildtype controls (**A, B, E, F**) or *Sp7Cre;Slc38a2^fl/fl^* or *Sp7Cre;Slc38a2^fl/+^* littermate controls (**C, D, G, H**) at embryonic day (E)15.5 (**A–D**) or P1 (**E–H**). Red arrow (**A–D**) or asterix (**E–H**) highlights reduced mineralization. A total of n = 7 or n = 5 *Slc38a2^LacZ/LacZ^* animals and n = 5 or n = 5 for *Sp7Cre;Slc38a2^fl/fl^* animals were analyzed at E15.5 or postnatal day (P)1, respectively. Scale bar = 5 mm. (**I–R**) Representative von Kossa staining (**I, J**), alkaline phosphatase (ALPL) staining (**K, L**) in situ hybridization for *Spp1*

*Figure 5 continued on next page*

*Figure 5 continued*

(**M, N**)*, Ibsp* (**O, P**), *Bglap* (**Q, R**), and *Col1a1* (**U, V**), or immunofluorescent staining for OSX (**S', S", T', T"**) and COL1A1 (**W, X**) on *Sp7Cre;Slc38a2*$^{fl/fl}$ (n = 4) (**J, L, N, P, R, T, V, X**) or *Sp7Cre;Slc38a2*$^{fl/+}$ (n = 4) (**I, K, M, O, Q, S, U, W**) newborn calvariae. *p≤0.05 by paired two-tailed Student's *t*-test. Scale bar = 100 μm.

The online version of this article includes the following source data and figure supplement(s) for figure 5:

**Figure supplement 1.** *Slc38a2* is required for osteoblast differentiation in vitro.

**Figure supplement 1—source data 1.** Numerical source data for ***Figure 5—figure supplement 1***.

**Figure supplement 2.** *Slc38a2*$^{LacZ/LacZ}$ mutants have impaired osteoblast differentiation during endochondral ossification.

**Figure supplement 2—source data 1.** Uncropped western blot source data for ***Figure 5—figure supplement 2***.

**Figure supplement 3.** *Sp7Cre;Slc38a2*$^{fl/fl}$ have impaired osteoblast differentiation during endochondral ossification.

**Figure supplement 3—source data 1.** Numerical and uncropped western blot source data for ***Figure 5—figure supplement 3***.

the consistent defect in intramembranous ossification in both genetic models, we focused our molecular analyses on the osteoblasts in the developing calvarium. *Sp7Cre;Slc38a2*$^{fl/fl}$ calvariae had normal alkaline phosphatase staining despite less mineralized area shown by von Kossa staining (***Figure 5I–L***). The defects in bone development are attributed to delayed osteoblast differentiation as *Sp7Cre;Slc38a2*$^{fl/fl}$ mice had significantly reduced expression of *Spp1*, *Ibsp*, and *Bglap* (***Figure 5M–R***). This was not due to a reduction in overall osteoblast numbers as there was no difference in the total number of *Sp7:GFP* expressing cells per mineralized area (***Figure 5S–T***). Despite this, significantly fewer *Sp7*:GFP-expressing cells were found to have OSX (encoded by *Sp7*) or RUNX2 protein expression in *Sp7Cre;Slc38a2*$^{fl/fl}$ animals (***Figure 5S-T***, ***Figure 5—figure supplement 3C***). Similar results were observed in the limbs of both *Slc38a2*$^{LacZ/LacZ}$ and *Sp7Cre;Slc38a2*$^{fl/fl}$ mice at E15.5 (***Figure 5—figure supplements 2B and 3D***). Similarly, *Sp7GFP*-expressing cells in *Sp7Cre;Slc38a2*$^{fl/fl}$ mice had significantly reduced COL1A1 protein expression despite normal *Col1a1* mRNA expression when compared to *Sp7Cre;Slc38a2*$^{fl/+}$ controls (***Figure 5U–X***, ***Figure 5—figure supplements 2B and 3C***). For comparison, the expression of proline-poor actin (as determined by phalloidin staining) and GFP (4.2% proline) were unaffected in *Sp7Cre;Slc38a2*$^{fl/fl}$ calvariae (***Figure 5—figure supplement 3C***). Collectively, these data indicate that *Slc38a2* provides proline essential for osteoblast differentiation and bone formation during bone development.

## Discussion

Here, we have defined a specific requirement for proline that arises during osteoblast differentiation and bone formation. Namely, that osteoblasts require proline to fulfill unique biosynthetic demands that arise due to increased production of proline-enriched osteoblast-associated proteins. To fulfill this demand, osteoblasts significantly increase proline consumption during differentiation. Genetically limiting proline uptake by ablating the proline transporter SLC38A2 results in delayed bone development. Mechanistically, osteoblasts utilize proline primarily for the synthesis of proline-enriched osteoblast proteins to facilitate both osteoblast differentiation and bone matrix production. Collectively, these data highlight a broad requirement for proline to regulate osteoblast differentiation and bone development in addition to supporting collagen synthesis.

Osteoblast differentiation is characterized by a distinct protein profile in addition to increasing bone matrix production (***Alves et al., 2010***; ***Zhang et al., 2007***). These osteoblast-associated proteins are enriched for the amino acid proline compared to all other proteins (***Figure 1***, ***Tables 1–2***). We and others have recently described increased consumption of numerous amino acids in differentiating osteoblasts including glutamine (***Sharma et al., 2021***; ***Stegen et al., 2021***; ***Yu et al., 2019***), asparagine (***Sharma et al., 2021***), and proline (this study). Glutamine and asparagine contribute to both de novo amino acid biosynthesis and protein synthesis directly (***Sharma et al., 2021***). In addition to

being directly incorporated into protein, proline can be oxidized in the inner mitochondrial membrane to form pyrroline-5-carboxylate (P5C) by proline dehydrogenase (PRODH). PRODH is a flavin dinucleotide (FAD)-dependent enzyme that donates electrons to complex II of the electron transport chain coupling proline oxidation to ATP synthesis (*Elia et al., 2017*; *Liu and Phang, 2012b*; *Olivares et al., 2017*; *Phang et al., 2012*). P5C can be converted back into proline by the NADPH-dependent enzyme pyrroline-5-carboxylate reductase (PYCR) in the proline cycle or can be converted into glutamate or other intermediate metabolites. Our tracing experiments did not find proline carbon enriched in either amino acids or trichloroacetic acid (TCA) intermediates. Due to technical reasons, we were not able to observe P5C in our experiments. For this reason, we conclude that proline is not widely metabolized past P5C in osteoblasts, but we are unable to make any conclusions about proline oxidation or the contribution of proline to bioenergetics in osteoblasts. Rather, our data indicates the primary use for proline is direct incorporation into nascent protein. Consistent with this, reducing proline availability specifically reduced the synthesis of proteins with higher-than-average proline content without affecting mTOR activation or inducing ISR (*Figures 3 and 4*). Proline depletion likely results in stalling of proline codons of proline-enriched genes at both the A and P sites of the ribosome similar to clear cell renal cell carcinoma that are vulnerable to proline depletion (*Loayza-Puch et al., 2016*). Consistent with this, protein expression was negatively correlated with the proline content in proteins when proline availability or uptake was limited (*Figures 3 and 4*). By comparison, limiting the availability of glutamine-induced robust activation of the ISR inhibits global protein synthesis (*Sharma et al., 2021*). This likely reflects the necessity of glutamine metabolism to maintain amino acid concentrations (including proline) and provide other metabolites during osteoblast differentiation (*Sharma et al., 2021*; *Stegen et al., 2021*; *Yu et al., 2019*). Consistent with the more direct use of proline in protein but not amino acid biosynthesis, we did not observe activation of the ISR in proline-free conditions despite reduced charging of proline tRNA. It is important to note that we evaluated the effects of proline withdrawal for 48 hr. This time point may miss the chronic effects of proline withdrawal as proline uptake is slow and the intracellular proline pool is stable with low turnover in naïve calvarial cells (*Figure 2*). Under these conditions, de novo biosynthesis of proline may be sufficient to meet the basal needs of naïve calvarial cells. Regardless, proline removal results in reduced synthetic efficiency of proline-rich proteins. This effect is likely exacerbated during osteoblast differentiation as these proline-rich proteins are increased.

Previous studies characterized proline uptake in both bones and osteoblasts directly. These studies described proline uptake occurring primarily via System A but did not identify individual transporters mediating proline uptake (*Adamson and Ingbar, 1967*; *Finerman and Rosenberg, 1966*; *Hahn et al., 1969*; *Yee, 1988*). Here, we identified the sodium-dependent neutral amino acid transporter SLC38A2 as responsible for approximately 55% of proline uptake in both calvarial osteoblasts and isolated bones (*Figure 4B*, *Figure 5—figure supplement 3B*). This is consistent with previous reports that System A mediates 60% of proline uptake in osteoblasts (*Yee, 1988*). Interestingly, SLC38A2 ablation affected only proline uptake (*Figure 5—figure supplement 3B*). It is not clear why SLC38A2 exclusively transports proline in osteoblasts as amino acid transporters are thought to be promiscuous in their substrate specificity (*Kandasamy et al., 2018*; *Teichmann et al., 2017*). For example, SLC38A2 is reported to transport alanine, serine, glycine, and glutamine in different cellular contexts (*Bröer et al., 2016*; *Morotti et al., 2019*). Our data indicates that glutamine is not a primary substrate for SLC38A2 in bone cells. This is consistent with our recent data demonstrating that glutamine uptake is mediated primarily by System ASC with no involvement of System A in osteoblasts (*Sharma et al., 2021*; *Shen et al., 2021*). In light of these data, a better understanding of the molecular regulation of SLC38A2 activity and substrate specificity is needed. In addition, it will be important to identify the transporters mediating SLC38A2-independent proline uptake as well as to understand their function during osteoblast differentiation and bone development.

Reducing proline uptake inhibited bone development in mice (*Figure 5*). This phenotype was attributed primarily to decreased osteoblast differentiation and reduced bone matrix production. Osteoblast differentiation and bone matrix production are associated with a unique biosynthetic demand for proline. Using a bioinformatic approach, we discovered that osteoblast-associated proteins are more enriched for proline than any other amino acid when compared to other cell types (*Figure 1*). Many of these proline-rich proteins are essential regulators of osteoblast differentiation (e.g., RUNX2, OSX, and ATF4), bone matrix production (e.g., COL1A1), or regulate the endocrine

functions of bone (e.g., OCN) (*Ducy et al., 1996*; *Ducy et al., 1997*; *Elefteriou et al., 2006*; *Kern et al., 2001*; *Nakashima et al., 2002*; *Otto et al., 1997*; *Yang et al., 2004*). Limiting proline availability by genetically ablating SLC38A2 specifically affected the production of proline-rich proteins (e.g., RUNX2, OSX, and COL1A1) in a manner that was proportional to the relative proline content. It is important to note that relatively minor reductions in protein expression or function are known to negatively impact osteoblast differentiation and bone development and underlie human bone diseases (*Baek et al., 2013*; *Bardai et al., 2016*; *Ben Amor et al., 2013*; *Choi et al., 2001*; *Lapunzina et al., 2010*; *Lee et al., 1997*; *Lou et al., 2009*; *Mundlos et al., 1997*; *Zhang et al., 2009*). Thus, ablating SLC38A2-dependent proline uptake has broad effects on osteoblast differentiation due to minor reductions in many proline-rich osteoblast regulatory proteins. This highlights an unappreciated mechanism by which osteoblast differentiation is vulnerable to nutrient (e.g., proline) limitations. When proline is available, osteoblast progenitors efficiently synthesize the proline-rich proteins necessary for differentiation (RUNX2 and OSX) and bone matrix deposition (COL1A1). The high proline content of these proteins presents a novel cellular checkpoint to ascertain if appropriate resources, in this case proline, are available for osteoblast differentiation to proceed. When proline is limited, these proteins are not efficiently synthesized, which limits osteoblast differentiation and bone matrix production until sufficient proline is available. This is critical to ensure that cells can meet the synthetic challenges associated with osteoblast differentiation and bone matrix production.

In addition to the aforementioned osteoblast defects, we also observed delayed COLX remodeling in *Sp7Cre;Slc38a2*$^{fl/fl}$ mice. This suggests that loss of SLC38A2 in hypertrophic chondrocytes contributes to delayed endochondral ossification in *Sp7Cre;Slc38a2*$^{fl/fl}$ mice. At this time, it is not apparent how SLC38A2 affects either chondrocyte differentiation or hypertrophy. It is intriguing to speculate that SLC38A2 provides chondrocytes with proline (or other amino acids) to facilitate the production of proline-rich collagens (e.g., COL2 and COLX) or other proteins during chondrogenesis and endochondral ossification. While the role of SLC38A2 in chondrogenesis was not a focus of this work, it will be important to determine both the necessity and the molecular substrates of SLC38A2 during chondrogenesis.

In summary, we have defined the necessity and the molecular substrates of *Slc38a2* in osteoblasts. Our data indicate that SLC38A2 acts cell-autonomously in osteoblasts to provide proline and that SLC38A2 is the major proline transporter in osteoblasts. Proline is essential for the production of proline-rich transcription factors (e.g., RUNX2 and OSX) and matrix proteins (COL1A1) necessary for osteoblast differentiation and bone formation. These data expand our understanding of the regulation of proline uptake and usage in osteoblasts and underscore the necessity of proline for osteoblast differentiation and bone development.

# Materials and methods

**Key resources table**

| Reagent type (species) or resource | Designation | Source or reference | Identifiers | Additional information |
|---|---|---|---|---|
| Genetic reagent (*Mus musculus*) | C57Bl/6J | Jackson Laboratory | RRID:IMSR_JAX:000664 | |
| Genetic reagent (*M. musculus*) | Rosa26Cas9 | Jackson Laboratory | RRID:IMSR_JAX:024858 | |
| Genetic reagent (*M. musculus*) | Rosa26Flpe | Jackson Laboratory | RRID:IMSR_JAX:003946 | |
| Genetic reagent (*M. musculus*) | Sp7tTA;tetOeGFP/Cre | PMID:16854976 | RRID:IMSR_JAX:006361 | |
| Genetic reagent (*M. musculus*) | Slc38a2LacZ | European Mouse Mutant Archive | | See 'Mouse strains' for more information |
| Chemical compound, drug | Ascorbic acid | Sigma | Cat# A4544 | |
| Chemical compound, drug | β-Glycerophosphate | Sigma | Cat# G9422 | |

*Continued on next page*

*Continued*

| Reagent type (species) or resource | Designation | Source or reference | Identifiers | Additional information |
|---|---|---|---|---|
| Chemical compound, drug | One-step NBT/BCIP solution | Thermo Fisher | Cat# PI34042 | |
| Chemical compound, drug | EasyTag EXPRESS S35 | PerkinElmer | Cat# NEG772002MC | |
| Chemical compound, drug | L-(3,4-$^3$H)-Glutamine | PerkinElmer | Cat# NET551250UC | |
| Chemical compound, drug | L-[1,2-$^{14}$C]-Alanine | PerkinElmer | Cat# NEC266E050UC | |
| Chemical compound, drug | L-(2,3-$^3$H)-Alanine | PerkinElmer | Cat# NET348250UC | |
| Chemical compound, drug | L-(2,3,4-$^3$H)-Proline | PerkinElmer | Cat# NET323250UC | |
| Chemical compound, drug | L-[$^3$H(G)]-Serine | PerkinElmer | Cat# NET248250UC | |
| Chemical compound, drug | L-[$^{14}$C(U)]-Glycine | PerkinElmer | Cat# NEC276E050UC | |
| Chemical compound, drug | L-[3,4-$^3$H]-Glutamate | PerkinElmer | Cat# NET490001MC | |
| Chemical compound, drug | Ultima Gold scintillation cocktail | PerkinElmer | Cat# 6013329 | |
| Chemical compound, drug | [U-$^{13}$C]-Glutamine | Sigma | Cat# 605166 | Used at 2 mM final concentration |
| Chemical compound, drug | [U-$^{13}$C]-Proline | Cambridge | Cat# 201740-83-2 | Used at 0.34 mM final concentration |
| Chemical compound, drug | [1,2-$^{13}$C]-Glucose | Sigma | Cat# 453188 | Used at 5.6 mM final concentration |
| Chemical compound, drug | AP substrate BM purple | Roche | Cat# 11442074001 | |
| Chemical compound, drug | ECL substrate | Bio-Rad | Cat# 1705060 | |
| Chemical compound, drug | Super signal West Femto ECL | Thermo Fisher | Cat# 1705060 | |
| Antibody | Eif2α (rabbit monoclonal) | Cell Signaling | RRID:AB_10692650 | (1:1000) |
| Antibody | pSer51 Eif2α (rabbit monoclonal) | Cell Signaling | RRID:AB_2096481 | (1:1000) |
| Antibody | pSer240/244 S6rp (rabbit polyclonal) | Cell Signaling | RRID:AB_331682 | (1:1000) |
| Antibody | S6rp (rabbit monoclonal) | Cell Signaling | RRID:AB_331355 | (1:1000) |
| Antibody | α-Tubulin (rabbit monoclonal) | Cell Signaling | RRID:AB_2619646 | (1:1000) |
| Antibody | β-Actin (rabbit polyclonal) | Cell Signaling | RRID:AB_330288 | (1:1000) |
| Antibody | HRP goat anti-rabbit (goat polyclonal) | Cell Signaling | RRID:AB_2099233 | (1:2000) |
| Antibody | HRP anti-mouse (horse polyclonal) | Cell Signaling | RRID:AB_330924 | (1:2000) |
| Antibody | Runx2 (rabbit monoclonal) | Cell Signaling | RRID:AB_10949892 | (1:1000) |
| Antibody | Smad1 (rabbit polyclonal) | Cell Signaling | RRID:AB_2107780 | (1:1000) |
| Antibody | 4E-BP1 (rabbit monoclonal) | Cell Signaling | RRID:AB_2097841 | (1:1000) |
| Antibody | mTOR (rabbit monoclonal) | Cell Signaling | RRID:AB_2105622 | (1:1000) |
| Antibody | Erk (rabbit monoclonal) | Cell Signaling | RRID:AB_390779 | (1:1000) |

*Continued on next page*

*Continued*

| Reagent type (species) or resource | Designation | Source or reference | Identifiers | Additional information |
|---|---|---|---|---|
| Antibody | eEF2 (rabbit polyclonal) | Cell Signaling | RRID:AB_10693546 | (1:1000) |
| Antibody | Phgdh (rabbit polyclonal) | Cell Signaling | RRID:AB_2750870 | (1:1000) |
| Antibody | Akt (rabbit polyclonal) | Cell Signaling | RRID:AB_329827 | (1:1000) |
| Antibody | COL1A1 (mouse monoclonal) | Santa Cruz | RRID:AB_2797597 | (1:1000) WB (1:200) IF |
| Antibody | OSX (mouse monoclonal) | Santa Cruz | RRID:AB_2895257 | (1:1000) WB |
| Antibody | ATF4 (rabbit polyclonal) | Santa Cruz | RRID:AB_2058752 | (1:1000) |
| Antibody | ATF2 (mouse monoclonal) | Santa Cruz | RRID:AB_626708 | (1:1000) |
| Antibody | PAX1 (mouse monoclonal) | Millipore | Cat# MABE1115 | (1:1000) |
| Antibody | SNAT2 (rabbit polyclonal) | Abcam | RRID:AB_2050321 | (1:1000) |
| Antibody | OSX (rabbit polyclonal) | Abcam | RRID:AB_2194492 | (1:200) IF |
| Antibody | COLX (mouse monoclonal) | Quartett | Cat# 2031501217 | (1:200) |
| Antibody | Goat anti-mouse 568 (goat unknown clonality) | Thermo Fisher | RRID:AB_141359 | (1:200) |
| Antibody | Goat anti-rabbit 568 (goat polyclonal) | Thermo Fisher | RRID:AB_143157 | (1:200) |
| Commercial assay or kit | Alexa Fluor 647 Phalloidin | Thermo Fisher | Cat# 22287 | |
| Commercial assay or kit | Iscript Reverse transcription kit | Bio-Rad | Cat# 1708841 | |
| Commercial assay or kit | SYBR green | Bio-Rad | Cat# 1725275 | |
| Commercial assay or kit | Click-iT EdU Alexa Fluor 488 Flow Cytometry Assay Kit | Invitrogen | Cat# C10420 | |
| Commercial assay or kit | Apoptosis Assay Kit (Cat# 22837) | AAT BIO | Cat# 22837 | |
| Chemical compound, drug | AP substrate BM purple | Roche | Cat# 11442074001 | |
| Software, algorithm | GraphPad 6 | https://www.graphpad.com/ | | |
| Software, algorithm | R version 3.6.0 | https://www.r-project.org/ | | |

## Mouse strains

C57Bl/6J (RRID:IMSR_JAX:000664), *Rosa26*^Cas9 (RRID:IMSR_JAX:024858), *Rosa26*^FLP (RRID:IMSR_JAX:003946), and *Sp7-tTA,tetO-EGFP/Cre* (RRID:IMSR_JAX:006361) mouse strains were obtained from the Jackson Laboratory. *Slc38a2*^LacZ (C57BL/6N-A<tm1Brd>Slc38a2<tm1a(KOMP)Wtsi>/Wtsi Ph) was purchased from the European Mouse Mutant Archive (https://www.emmanet.org). To generate *Slc38a2*^flox, *Slc38a2*^LacZ mice were crossed to *Rosa26*^FLP to remove FRT-flanking LacZ cassette followed by backcrossing with C57Bl/6J to remove *Rosa26*^FLP allele. Mice were housed at 23°C on a 12 hr light/dark cycle with free access to water and PicoLab Rodent Diet 20 (LabDiet #5053, St. Louis, MO). All mouse procedures were approved by the Animal Studies Committees at Duke University first and then the University of Texas Southwestern Medical Center at Dallas (Animal Protocol 2020-102999).

## Mouse analyses

Skeletal preparations were performed on E15.5 or P0 embryos obtained from timed pregnancies. Noon of the day of plugging was considered 12 hr post coitum or E0.5. Embryos were dehydrated in 95% ethanol overnight followed by submersion in acetone overnight. Specimens were then stained with 0.03% Alcian blue in 70% ethanol and 0.005% Alizarin red in water overnight. Stained embryos

were then cleared in 1% KOH prior to a graded glycerol series (30, 50, and 80%). For histological analyses, freshly isolated limbs or calvariae were fixed in 4% PFA at 4°C overnight. Limbs were then processed and embedded in paraffin and sectioned at 5 µm using a Microtome (Leica RM2255). Calvariae were cryoprotected in 30% sucrose overnight, embedded in OCT, and sectioned at 10µm using a Cryostat (Leica CM1950).

## In situ hybridization

In situ hybridization was performed on 10 µm cryosectioned calvariae or 5 µm paraffin-sectioned limbs. Cryosections were washed with water first for 5 min. Paraffin sections were deparaffinized and rehydrated, followed by 20 mg/mL proteinase K treatment for 10 min. Sections were first then fixed in 4% PFA for 10 min followed by 10 min acetylation. Sections were then incubated in hybridization buffer for 2 hr at room temperature. Digoxigenin-labeled antisense RNA probes for *Col1a1* (HindIII, T7), *Sp7* (NotI, T3), *Spp1* (EcoRI, SP6), *Ibsp* (NOTI, SP6), or *Bglap* (XbaI, T3) were hybridized at 60°C overnight.

## Immunohistochemistry

Sections were blocked in 1.5% goat serum in PBST and incubated with the following primary antibodies (1:250 in blocking solution) as indicated: anti-Col1a (AB_1672342), anti-Osx (AB_2194492), anti-Runx2 (AB_10949892), or anti-COLX (Quartett, 2031501005) at 4°C overnight. Sections were then incubated with Alexa Fluor 568 goat anti-rabbit (AB_143157)/-mouse IgG(H+L) antibody (AB_2534072) at 1:250 dilution at room temperature for 30 min. Sections were post-fixed in 4% PFA for 10 min before mounting. For actin staining, Alexa Fluor 647 Phalloidin (Invitrogen; 1:200 in blocking buffer) was applied to sections before mounting. Sections were mounted using Heatshield with DAPI (Vector). For COLX IF staining, antigen retrieval was performed by incubating sections in 0.4 mg/mL pepsin (0.01 N HCl) at 37°C for 10 min.

## Cell culture

Primary calvarial osteoblasts were isolated as follows. The calvaria of P4 pups was harvested and extemporaneous tissue was removed. The calvariae were chopped with a scissor into small pieces and washed with PBS twice. The calvaria pieces were then incubated in 1.8 mg/mL Collagenase P in PBS for 10 min with agitation at 37°C four times. The first digestion was discarded, and the last three digestions were collected and run through 70 µm cell strainer. Cells were then centrifuged at $350 \times g$ for 5 min and cultured in T75 flasks in aMEM containing 15% FBS at 37°C and 5% $CO_2$. Cells were plated at $1 \times 10^5$ cells/mL for further experiments when they reached 90% confluency. Osteoblast differentiation was induced at 100% confluency using aMEM supplemented with 50 mg/mL ascorbic acid and 10 mM β-glycerophosphate for the indicated time with a change of media every 48 hr. For proline dropout experiments, primary calvarial cells were treated with proline-free aMEM (Genaxxon) supplemented back to 0.3 mM proline or not for the indicated length of time. To evaluate the synthesis of individual proteins, CHX washout experiments were performed. Calvarial osteoblasts were treated with 10 µg/mL CHX for 24 hr. Cells were then chased with aMEM containing either 0.3 mM or 0 mM proline for up to 24 hr before proteins were harvested. Alkaline phosphatase activity was assessed using 5-bromo-4-chloro-3′-indolyphosphate/nitro blue tetrazolium (BCIP/NPT). Mineralization was visualized by either von Kossa or Alizarin red staining as indicated.

## CRISPR/Cas9 targeting

Lentiviral vectors expressing single-guide RNAs (sgRNA) targeting either *Slc38a2* or Luciferase and mCherry were cloned into the LentiGuide-Puro plasmid according to the previously published protocol (*Sanjana et al., 2014*). The LentiGuide-Puro plasmid was a gift from Feng Zhang (Addgene plasmid #52963). Sequences of each sgRNA protospacer are shown in *Supplementary file 1*. To make viral particles, the sgRNA carrying lentiviral vector was cotransfected in 293T cells with the plasmids pMD2.g and psPax2. Virus containing media was collected and run through 0.45 µm filter. Calvarial osteoblasts harvested from *Rosa26^{Cas9/Cas9}* pups were infected for 24 hr and recovered for 24 hr in regular media before further experiments.

## Mass spectrometry

Calvarial osteoblasts were cultured in 6 cm plates until confluency before sample preparation for mass spectrometry. For glucose, glutamine, and proline-tracing experiments, naïve or differentiated

calvarial cells were cultured in aMEM (Genaxxon) containing 0.3 mM [U-$^{13}$C]-proline (Sigma-Aldrich), 2 mM [U-$^{13}$C]-glutamine (Cambridge) or 5.6 mM [1,2-$^{13}$C]-glucose (Sigma-Aldrich) for 24 hr or 72 hr. The labeling was terminated with ice-cold PBS, and cells were scrapped with –20°C 80% methanol on dry ice. 20 nmol norvaline was added into each methanol extract as internal control, followed by centrifugation at 10,000 × $g$ for 15 min. Supernatants were processed and analyzed by the Metabolomics Facility at the Children's Medical Center Research Institute at UT Southwestern. For tracing experiments into protein, cells were labeled for 0, 12, 24, or 72 hr. Cells were then scrapped in 1 M perchloric acid. The protein pellet was washed with 70% ethanol three times. The pellet was then incubated with 1 mL of 6 M HCl at 110°C for 18 hr to hydrolyze the proteins. 1 mL of chloroform was then added to each sample followed by centrifugation at 400 × $g$ for 10 min. Supernatants were taken for further preparation. The supernatant was dried by N$_2$ gas at 37°C. GC-MS method for small polar metabolites assay used in this study was adapted from *Wang et al., 2018*. The dried residues were resuspended in 25 µL methoxylamine hydrochloride (2% [w/v] in pyridine) and incubated at 40°C for 90 min. 35 µL of MTBSTFA + 1% TBDMS was then added, followed by 30 min incubation at 60°C. The supernatants from proline-tracing experiments were dried by N$_2$ gas at 37°C followed by resuspension in 50 µL of MTBSTFA + 1% TBDMS incubated at 60°C for 30 min. The derivatized sampled were centrifuged for 5 min at 10,000 × $g$ force. Supernatant from each sample was transferred to GC vials for analysis. 1 µL of each sample was injected in split or splitless mode depending on the analyte of interest. GC oven temperature was set at 80°C for 2 min, increased to 280°C at a rate of 7°C/min, and then kept at 280°C for a total run time of 40 min.

GC-MS analysis was performed on an Agilent 7890B GC system equipped with an HP-5MS capillary column (30 m, 0.25 mm i.d., 0.25 mm-phase thickness; Agilent J&W Scientific), connected to an Agilent 5977A mass spectrometer operating under ionization by electron impact (Meister, 1975) at 70 eV. Helium flow was maintained at 1 mL/min. The source temperature was maintained at 230°C, the MS quad temperature at 150°C, the interface temperature at 280°C, and the inlet temperature at 250°C. Mass spectra were recorded in selected ion monitoring (SIM) mode with 4 ms dwell time.

## Amino acid uptake assay

Amino acid uptake assays were performed as previously described (*Shen and Karner, 2021*). Cells were first washed three times with PBS and incubated with Krebs–Ringer HEPES (KRH) (120 mM NaCl, 5 mM KCl, 2 mM CaCl$_2$, 1 mM MgCl$_2$, 25 mM NaHCO$_3$, 5 mM HEPES, 1 mM D-glucose) with 4 µCi/mL L-[2,3,4-$^3$H]-proline (PerkinElmer NET323250UC), L-[3,4-$^3$H]-glutamine (PerkinElmer NET551250UC), L-[2,3-$^3$H]-alanine (PerkinElmer NET348250UC), L-[1,2-$^{14}$C]-alanine (PerkinElmer NEC266E050UC), L-[$^3$H(G)]-serine (PerkinElmer NET248250UC), L-[$^{14}$C(U)]-glycine (PerkinElmer NEC276E050UC), or L-[3,4-$^3$H]-glutamate (PerkinElmer NET490001MC) for 5 min at 37°C. Uptake and metabolism were terminated with ice-cold KRH, and the cells were scraped with 1% SDS. Cell lysates were combined with 8 mL Ultima Gold scintillation cocktail (PerkinElmer 6013329), and Counts per minutes (CPM) was measured using Beckman LS6500 Scintillation counter. Newborn mouse humeri and femurs were used for ex vivo amino acid uptake acid. Extemporaneous and cartilaginous tissues were removed from the bones, and counter lateral parts were harvested and boiled for normalization. Bones were then incubated with KRH containing radiolabeled amino acids for 30 min at 37°C. Uptake and metabolism were terminated by ice-cold KRH. Samples were homogenized in RIPA lysis buffer (50 mM Tris [pH 7.4], 15 mM NaCl, 0.5% NP-40, 0.1% SDS, 0.1% sodium deoxycholate) followed by sonication using an Ultrasonic Processor (VCX130) (amplitude: 35%; pulse 1 s; duration: 10 s) and centrifugation. Supernatant from each sample was combined with 8 mL scintillation cocktail, and CPM was measured using Beckman LS6500 Scintillation counter. Radioactivity was normalized with the boiled contralateral bones.

## Proline incorporation assay

Cells were incubated with KRH supplemented with 4 µCi/mL L-[2,3,4-$^3$H]-proline for 3 hr. Cells were lysed with RIPA and followed by centrifugation. Protein was precipitated with TCA and resuspended using 1 mL 1 M NaOH. 200 µL of the dissolved sample was saved for radioactivity reading later as the total proteins. The rest of each sample was split into two: one was treated with 15 mg Collagenase P and 60 mM HEPES to digest collagens and the other with only 60 mM HEPES as the baseline control. Samples were incubated at 37°C for 3 hr. After incubation, residual proteins and Collagenase P were

precipitated using TCA followed by centrifugation. Supernatant from each sample was combined with 8 mL scintillation cocktail, and CPM was measured using Beckman LS6500 Scintillation counter. Radioactivity for collagen incorporation was normalized with 60 mM HEPES treated as the baseline control.

## Metabolic labeling with S$^{35}$-cysteine/methionine

Cells were incubated with cysteine/methionine-free DMEM supplemented with 165 mCi of EasyTag EXPRESS S$^{35}$ protein labeling mix (PerkinElmer) for 30 min. Cells were then lysed with RIPA buffer followed by centrifugation. Lysates were spotted on Whatman paper. Protein was precipitated with cold 5% TCA and washed with 10% TCA, ethanol, and acetone. The Whatman paper was air dried for 10 min and dipped into 8 mL scintillation cocktail. Radioactivity was measured using LS6500 Scintillation counter and normalized with cell number.

## RNA isolation and qPCR

Total RNA was harvested from calvarial osteoblasts using TRIzol and purified by mixing with chloroform. 500 ng of total RNA was used for reverse transcription by IScript cDNA synthesis kit (Bio-Rad). SsoAdvanced Universal SYBR Green Supermix (Bio-Rad) was used for qPCR with primers used at 0.1 µM (listed *Supplementary file 3*). Technical and biological triplicates were performed using a 96-well plate on an ABI QuantStudio 3. The PCR program was set as 95°C for 3 min followed by 40 cycles of 95°C for 10 s and 60°C for 30 s. *Actb* mRNA level was used to normalize the expression of genes of interest, and relative expression was calculated using the 2$^{-(\Delta\Delta Ct)}$ method. PCR efficiency was optimized, and melting curve analyses of products were performed to ensure reaction specificity.

## RNAseq

RNA sequencing was performed in biological triplicate by the Duke University Center for Genomic and Computational Biology Sequencing and Genomic Technology Shared Resource on 10 mg of RNA isolated from primary calvarial cells cultured in either growth or osteogenic media for 7 days. RNAseq data was processed using the TrimGalore toolkit, which employs Cutadapt to trim low-quality bases and Illumina sequencing adapters from the 3′ end of the reads. Only reads that were 20 nt or longer after trimming were kept for further analysis. Reads were mapped to the GRCm38v68 version of the mouse genome and transcriptome using the STAR RNA-seq alignment tool. Reads were kept for subsequent analysis if they mapped to a single genomic location. Gene counts were compiled using the HTSeq tool. Only genes that had at least 10 reads in any given library were used in subsequent analysis. Normalization and differential expression were carried out using the DESeq2 Bioconductor package with the R statistical programming environment. The false discovery rate was calculated to control for multiple hypothesis testing. Gene set enrichment analysis9 was performed to identify differentially regulated pathways and GO terms for each of the comparisons performed.

## Western blotting

Calvarial osteoblasts were scraped in RIPA lysis buffer with cOmplete Protease Inhibitor and PhosSTOP cocktail tablets (Roche). Protein concentration was determined by BCA protein assay kit (Thermo). Protein (6–20 µg) was loaded on 4–15% or 12% polyacrylamide gel and transferred onto Immuno-Blot PVDF membrane. The membranes were blocked for 1 hr at room temperature in 5% milk powder in TBS with 0.1% Tween (TBST) and then incubated at 4°C with the primary antibody overnight. Primary antibodies were used at 1:1000 to detect proteins, listed as follows: anti-SNAT2 (AB_2050321), anti-P-S240/244S6 (AB_331682), anti-S6 (AB_331355), anti-P-S51 Eif2a (AB_2096481), anti-Eif2a (AB_10692650), anti-Col1a1 (AB_1672342), anti-Runx2 (AB_10949892), anti-β-actin (AB_330288), anti-Smad1 (AB_2107780), anti-4E-BP1 (AB_2097841), anti-ATF4 (AB_2058752), anti-ATF2 (AB_626708), anti-mTOR (AB_2105622), anti-Akt (AB_329827), anti-Erk (AB_390779), anti-eEF2 (AB_10693546), anti-Phgdh (AB_2750870), anti-α-tubulin (AB_2619646), anti-Osx (AB_2895257), and PAX1 (Millipore MABE1115). Membranes were then incubated at room temperature with anti-rabbit IgG (AB_2099233) or anti-mouse IgG, HRP-linked antibody (AB_330924) at 1:2000 for 1 hr at room temperature. Immunoblots were next developed by enhanced chemiluminescence (Clarity Substrate Kit or SuperSignal West Femto substrate). Each experiment was repeated with at least three independently prepared protein extractions. Densitometry was performed for quantification for each blot.

## Amino acid proportion and amino acid demand prediction analysis

Amino acid sequences of proteins (Mus_musculus.GRCm38.pep.all.fa) were retrieved from Ensembl (https://uswest.ensembl.org/info/data/ftp/index.html). Amino acid proportion was calculated based on the amino acid sequences of specific proteins (RUNX2, COL1A1, OSX, and OCN) and proteins associated with different GO terms. mRNA expression of genes in undifferentiated and differentiated osteoblasts was obtained from transcriptomic analysis. Top 500 induced and suppressed genes from differentiated osteoblasts were selected for the calculation of proline proportion. For amino acid demand prediction, amino acid proportion and mRNA expression were merged using *Gene.stable. ID* as the bridge. 75 unmatched proteins were excluded from a total of 49,665 proteins. To predict the amino acid demand change, changes in mRNA expression were assumed to be proportional to changes in protein translation. Based on this, the change of amino acid demand in each protein is proportional to mRNA expression change:

$$\Delta AA \propto \Delta R \times N_{aa}$$
$$AA = \text{amino acid demand}$$
$$R = \text{mRNA abundance}$$
$$N_{aa} = \text{number of amino acids}$$

To summarize the overall change of amino acid demand during osteoblast differentiation:

$$\%\Delta AA = \frac{\sum \left[ \left( R_{differentiated} - R_{undifferentiated} \right) \times N_{aa} \right]}{\sum \left[ R_{undifferentiated} \times N_{aa} \right]} \times 100\%$$

## tRNA aminoacylation assay

The method was adapted from *Loayza-Puch et al., 2016* and *Saikia et al., 2016*. Purified RNA was resuspended in 30 mM NaOAc/HOAc (pH 4.5). RNA was divided into two parts (2 µg each): one was oxidized with 50 mM NaIO$_4$ in 100 mM NaOAc/HOAc (pH 4.5) and the other was treated with 50 mM NaCl in NaOAc/HOAc (pH 4.5) for 15 min at room temperature. Samples were quenched with 100 mM glucose for 5 min at room temperature, followed by desalting using G50 columns and precipitation using ethanol. tRNA was then deacylated in 50 mM Tris-HCl (pH 9) for 30 min at 37°C, followed by another ethanol precipitation. RNA (400 ng) was then ligated the 3′ adaptor (5′-/5rApp/TGGA ATTCTCGGGTGCCAAGG/3ddC/-3′) using T4 RNA ligase 2 (NEB) for 4 hr at 37°C. 1 µg RNA was then reverse transcribed using SuperScript III first-strand synthesis system with the primer (GCCTTGGC ACCCGAGAATTCCA) following the manufacturer's instruction. Relative charging level was calculated by qRT-PCR using tRNA-specific primers stated in *Supplementary file 2*.

## Flow cytometry

Flow cytometry was used to analyze 5-ethynyl-2′-deoxyuridine (EdU) incorporation and cell viability in calvarial osteoblasts. EdU incorporation was performed using Click-iT EdU Alexa Fluor 488 Flow Cytometry Assay Kit. Cells were incubated with EdU (10 µM) for 24 hr. Cells were then trypsinized, fixed, permeabilized, and incubated with Click-iT reaction cocktail for 30 min according to the manufacturer's instructions. Cell viability was analyzed using the Cell Meter APC-Annexin V Binding Apoptosis Assay Kit (Cat# 22837). Cells were trypsinized and incubated with APC-Annexin V conjugate and propidium iodide for 30 min. Cells were all resuspended in 500 µL PBS and analyzed using FACSCanto II flow cytometer (BD Biosciences). Data were analyzed and evaluated using FlowJo (v.11).

## Quantification and statistical analysis

Statistical analyses were performed using either GraphPad Prism 6 or R software. One-way ANOVA or unpaired two-tailed Student's *t*-test were used to determine statistical significance as indicated in the text. All data are shown as mean values ± SD or SEM as indicated. $p < 0.05$ is considered to be statistically significant. Sample size (n) and other statistical parameters are included in the figure legends. Experiments were repeated on a minimum of three independent samples unless otherwise noted.

# Acknowledgements

We thank Drs. Thomas Carroll and Guoli Hu for critical comments on this manuscript. We also thank Jessica Sudderth and the Children's Medical Center Metabolomics Core Facility for analysis and

interpretation of GC/MS data. This work was supported by the National Institute of Arthritis and Musculoskeletal and Skin Diseases (NIAMS) grants (AR076325 and AR071967) to CMK.

## Additional information

### Funding

| Funder | Grant reference number | Author |
| --- | --- | --- |
| National Institute of Arthritis and Musculoskeletal and Skin Diseases | AR071967 | Courtney M Karner |
| National Institute of Arthritis and Musculoskeletal and Skin Diseases | AR076325 | Courtney M Karner |

The funders had no role in study design, data collection and interpretation, or the decision to submit the work for publication.

### Author contributions

Leyao Shen, Investigation, Writing - original draft; Yilin Yu, Yunji Zhou, Shondra M Pruett-Miller, Investigation; Guo-Fang Zhang, Investigation, Writing - review and editing; Courtney M Karner, Conceptualization, Investigation, Supervision, Writing - review and editing

### Author ORCIDs

Leyao Shen http://orcid.org/0000-0002-4952-437X
Shondra M Pruett-Miller http://orcid.org/0000-0002-3793-585X
Courtney M Karner http://orcid.org/0000-0003-0387-4486

### Ethics

All mouse procedures were approved by the Animal Studies Committees at Duke University first and then the University of Texas Southwestern Medical Center at Dallas (Animal Protocol 2020-102999).

### Decision letter and Author response

Decision letter https://doi.org/10.7554/eLife.76963.sa1
Author response https://doi.org/10.7554/eLife.76963.sa2

## Additional files

### Supplementary files

- Supplementary file 1. sgRNA protospacer sequence.
- Supplementary file 2. RT-PCR primer sequences for tRNA charging.
- Supplementary file 3. RT-PCR primer sequences.
- Transparent reporting form

### Data availability

All data generated or analyzed during this study are included in this submission and the supporting files. Source data files are included for all western blot images and excel spreadsheets are included for the RNAseq and metabolic tracing experiments in figures 1 and 2.

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
