## [Editor Report]

The article is novel and informative; the authors' conclusions are supported by the data as shown. The article is significant as it proves that a key function of proline during bone formation is being incorporated into proline-enriched proteins rather than contributing to other metabolic processes.

---

## [Decision Letter]

**Decision letter after peer review:**

Thank you for submitting your article "SLC38A2 provides proline to fulfilll unique synthetic demands arising during osteoblast differentiation and bone formation." for consideration by *eLife*. Your article has been reviewed by 3 peer reviewers, one of whom is a member of our Board of Reviewing Editors, and the evaluation has been overseen by Carlos Isales as the Senior Editor. The reviewers have opted to remain anonymous.

The data are outstanding and supportive of the authors conclusions. However, reviewers have comments and concerns that need to be addressed point-by-point and some of them may require additional experiments.

In particular:

1. Document whether the process of endochondral bone development is delayed in mutant mice.

2. Analysis of de novo proline synthesis upon proline withdrawal and/or SLC38A2 inactivation?

3. Proper statistical analysis of the data.

*Reviewer #1 (Recommendations for the authors):*

Since OSX-Cre is expressed in hypertrophic chondrocytes, it would be helpful to better document whether mutant mice lacking SCL38A2 in OSX-Cre lineage cells experience a delay of replacement of cartilage by bone and thus of endochondral bone development as such a delay could contribute to the impaired bone formation observed in those mice.

*Reviewer #2 (Recommendations for the authors):*

Shen et al., have investigated the role of proline and its transporter SLC38A2 in osteoblasts (OB). OB marker proteins and regulators are enriched in proline. In a series of logical and well-designed experiments, they show that depletion of proline from OBs in vitro impairs their differentiation and function while genetic deletion of SLC38A2 in vivo, disrupts bone formation.

They show that proline is mostly transported from outside and not much is synthesized in OB. This is why studying proline transporter is important for this work. The study design is elegant and involves mechanistic aspect of the role of proline and SLC38A2 in OB function and bone formation. Experiments include appropriate controls in vitro and in vivo. Statistical methods are appropriate but n is not always specified and missing in some graphs. It would also be more rigorous if the Authors used scatter plots along with bars to show individual values rather than Mean and SD.

*Reviewer #3 (Recommendations for the authors):*

1. SLC38A2/SNAT2 is not only a proline transporter, but also facilitates uptake of other amino acids such as glutamine, serine, glycine and alanine, which are known to affect osteoblast function. The authors show that pharmacological/genetic SLC38A2 inactivation does not affect their uptake, but is this also reflected in intracellular glutamine, serine, glycine and alanine levels?

2. de novo proline synthesis is low in baseline conditions, is this altered upon proline withdrawal and/or SLC38A2 inactivation?

3. In Figure 2D, the authors show that relative proline incorporation in both total protein and collagen is increased when cells are cultured in osteogenic differentiation medium. How are the absolute values of protein incorporation? Is the demand for collagen higher?

4. Does proline withdrawal/SLC38A2 inactivation affect total protein synthesis and/or protein content? This is especially relevant since the levels of EIF4EBP1, an important regulator of protein translation, are decreased.

5. For mechanistic insight, the authors rely on adequately executed experiments using a pharmacological inhibitor of SLC38A2 and CRISPR-Cas9 mediated knockdown. To clearly link their in vitro findings with their in vivo model, these experiments using osteoblasts isolated from Sp7Cre;Slc38a2fl/fl mice.

6. Scale bars are missing in Figure 5, and Figure 5 supplement 2 and 3.

---

## [Author Response]

The data are outstanding and supportive of the authors conclusions. However, reviewers have comments and concerns that need to be addressed point-by-point and some of them may require additional experiments.In particular:1. Document whether the process of endochondral bone development is delayed in mutant mice.2. Analysis of de novo proline synthesis upon proline withdrawal and/or SLC38A2 inactivation?3. Proper statistical analysis of the data.

We thank the reviewers for the positive review of our work. Moreover, we appreciate the opportunity to improve our manuscript. To address these particular points, we have done the following: We evaluated COLX protein expression at e15.5 and can confirm that endochondral ossification is delayed in the *Sp7Cre;Slc38a2^fl/fl^* mice. We have included this new data in Figure 5 Supplement 3D and updated the discussion. We also evaluated de novo proline synthesis and found proline synthesis from glutamine increases significantly when *Slc38a2* is ablated. These data are included in Figure 4 Figure Supplement 1D. Finally, we replaced all the original bar graphs with graphs showing all data points. For clarity, we have highlighted all changes to the text of the manuscript using red font. Specific comments to the individual comments are found below.

Reviewer #1 (Recommendations for the authors):Since OSX-Cre is expressed in hypertrophic chondrocytes, it would be helpful to better document whether mutant mice lacking SCL38A2 in OSX-Cre lineage cells experience a delay of replacement of cartilage by bone and thus of endochondral bone development as such a delay could contribute to the impaired bone formation observed in those mice.

We thank the reviewer for the very positive review of our work. Because *Sp7Cre* is expressed in hypertrophic chondrocytes we initially chose to highlight the calvaria phenotype which develops via intramembranous ossification without a cartilage intermediate. To address the reviewer’s concern, we evaluated COLX expression at e15.5. COLX removal was delayed indicating endochondral ossification is delayed in the *Sp7Cre;Slc38a2^fl/fl^* mice. We have included this new data in Figure 5 Supplement 3D. These data indicate that SLC38A2 ablation in chondrocytes likely contributes to the endochondral ossification defects observed in these mice. We have expanded the discussion to include the following paragraph:

“In addition to the aforementioned osteoblast defects, we also observed delayed COLX remodeling at e15.5 in *Sp7Cre;Slc38a2^fl/fl^* mice. This suggests that loss of SLC38A2 in hypertrophic chondrocytes contributes to delayed endochondral ossification in *Sp7Cre;Slc38a2^fl/fl^* mice. At this time, it is not apparent how SLC38A2 affects either chondrocyte differentiation or hypertrophy. It is intriguing to speculate that SLC38A2 provides chondrocytes with proline (or other amino acids) to facilitate the production of proline rich collagens (e.g. COL2 and COLX) or other proteins during chondrogenesis and endochondral ossification. While the role of SLC38A2 in chondrogenesis was not a focus of this work it will be important to determine both the necessity and the molecular substrates of SLC38A2 during chondrogenesis.”

Reviewer #2 (Recommendations for the authors):Shen et al., have investigated the role of proline and its transporter SLC38A2 in osteoblasts (OB). OB marker proteins and regulators are enriched in proline. In a series of logical and well-designed experiments, they show that depletion of proline from OBs in vitro impairs their differentiation and function while genetic deletion of SLC38A2 in vivo, disrupts bone formation.They show that proline is mostly transported from outside and not much is synthesized in OB. This is why studying proline transporter is important for this work. The study design is elegant and involves mechanistic aspect of the role of proline and SLC38A2 in OB function and bone formation. Experiments include appropriate controls in vitro and in vivo. Statistical methods are appropriate but n is not always specified and missing in some graphs. It would also be more rigorous if the Authors used scatter plots along with bars to show individual values rather than Mean and SD.

We appreciate the positive review of our work. We have updated all the figure legends to include the sample size for all experiments and we have modified all bar graphs to show the individual data points.

Reviewer #3 (Recommendations for the authors):1. SLC38A2/SNAT2 is not only a proline transporter, but also facilitates uptake of other amino acids such as glutamine, serine, glycine and alanine, which are known to affect osteoblast function. The authors show that pharmacological/genetic SLC38A2 inactivation does not affect their uptake, but is this also reflected in intracellular glutamine, serine, glycine and alanine levels?

The reviewer raises an important point about the specificity of SLC38A2 for proline. To evaluate intracellular amino acid levels, we performed mass spectrometry in the SLC38A2 knockout calvarial cells. Proline and glutamine were both significantly reduced whereas no other amino acid was significantly affected by loss of SLC38A2. The reduction in glutamine concentration was not due to reduced glutamine uptake but rather is attributed to increased de novo proline synthesis from glutamine. These data are now included in the new Figure 4 Figure Supplement 1C-D.

2. de novo proline synthesis is low in baseline conditions, is this altered upon proline withdrawal and/or SLC38A2 inactivation?

We observed a compensatory increase in de novo proline biosynthesis from ^13^C_U_-glutamine in SLC38A2 knockout cells.

3. In Figure 2D, the authors show that relative proline incorporation in both total protein and collagen is increased when cells are cultured in osteogenic differentiation medium. How are the absolute values of protein incorporation? Is the demand for collagen higher?

The protein samples for total protein and collagen incorporation are processed quite differently (see Materials and methods). Moreover, total protein contains collagenous proteins. Because of this, it is difficult to directly compare the absolute proline incorporation values. Regardless, the overall demand for proline for collagen significantly increases during osteoblast differentiation.

4. Does proline withdrawal/SLC38A2 inactivation affect total protein synthesis and/or protein content? This is especially relevant since the levels of EIF4EBP1, an important regulator of protein translation, are decreased.

To address this concern, we performed ^35^S-Cysteine/Methionine incorporation assay to measure protein synthesis directly. This experiment found that protein synthesis was significantly reduced in SLC38A2 knockout calvarial cells. There are many potential explanations for this decline. First, proline depletion directly affects the translation of proline rich proteins (Figure 3C). Second, collagen is one of the highest expressed proteins in osteoblasts and decreased collagen synthesis would be reflected in the overall rate of protein synthesis. Consistent with this, we observed decreased collagen synthesis in SLC38A2 knockout cells. Finally, as the reviewer alluded to, reduced EIF4EBP1 expression is predicted to reduce protein synthesis. In light of these possibilities, further investigation of the regulation of protein synthesis by proline in osteoblasts is warranted.

5. For mechanistic insight, the authors rely on adequately executed experiments using a pharmacological inhibitor of SLC38A2 and CRISPR-Cas9 mediated knockdown. To clearly link their in vitro findings with their in vivo model, these experiments using osteoblasts isolated from Sp7Cre;Slc38a2fl/fl mice.

We concur it is important to link the in vitro findings with the in vivo model. For this reason, we evaluated both mRNA and protein expression of the same proteins both in vitro *and* in the *Sp7Cre;Slc38a2^fl/fl^* mice using immunofluorescence techniques. These experiments confirmed the reduced protein expression of RUNX2, OSX and COL1A1 despite no changes in mRNA expression. This highlights the necessity of proline availability to regulate the synthesis of proline enriched proteins without affecting mRNA expression in osteoblasts and bones in vivo.

6. Scale bars are missing in Figure 5, and Figure 5 supplement 2 and 3.

We have added scale bars to Figure 5 and Figure 5 Supplemental 2 and 3